# Optimizing canine T cell activation, expansion, and transduction

Treyvon W. Davis[1], Jennifer C. Holmes[2], Arissa He[3], Paul R. Hess [2], Christopher L. Mariani[2], Yevgeny Brudno[4,5,6,7] *

1 Department of Biological Sciences, College of Veterinary Medicine, North Carolina State University, North Carolina, Raleigh, United States of America, 2 Department of Clinical Sciences, College of Veterinary Medicine, North Carolina State University, North Carolina, Raleigh, United States of America, 3 College of Agriculture and Life Sciences, North Carolina State University, North Carolina, Raleigh, United States of America, 4 Division of Pharmacoengineering and Molecular Pharmaceutics, Eshelman School of Pharmacy, University of North Carolina at Chapel Hill, Chapel Hill, North Carolina, United States of America, 5 Joint Department of Biomedical Engineering, University of North Carolina at Chapel Hill and North Carolina State University, Raleigh, North Carolina, United States of America, 6 Comparative Medicine Institute, North Carolina State University, Raleigh, North Carolina, United States of America, 7 Lineberger Comprehensive Cancer Center, University of North Carolina at Chapel Hill, Chapel Hill, North Carolina, United States of America

* ybrudno@email.unc.edu

## Abstract

Dogs are becoming an important model for human cancers, and successfully troubleshooting issues with genetically modified T cell immunotherapy for round cell and solid neoplasms in dogs provides a unique opportunity to improve efficacy, safety, and affordability for humans as well. Unfortunately, T cell activation in dogs for optimal viral transduction has not been determined, restricting advancements in canine T cell immunotherapy. Two αCD3 and two αCD28 antibody clones for canine T cell stimulation have been described in the literature, but no studies have been undertaken to evaluate which αCD3/αCD28 combination is most effective, nor has anyone directly compared the efficacy of the two most popular antibody presentation strategies: antibody-coated plates and antibody-conjugated beads. In evaluating the effects of plate- or bead-bound αCD3 stimulation alone versus αCD3/αCD28 in combination, we tested 12 possible antibody stimulation strategies in addition to evaluating two largely unexplored mitogens in canine T cell transduction, phorbol myristate acetate (PMA) with ionomycin and concanavalin A (ConA). We investigated the impact of these stimulation strategies on canine T cell activation, expansion, and transduction. For stimulation strategies producing the best results, we also examined how each strategy affected the proportions of CD4/CD8 T cell subsets and regulatory T cell ($T_{reg}$) prevalence. We determined that, in general, plate-bound antibodies were far superior to bead-bound antibodies for canine T cell stimulation, and that plate-bound αCD3 clone CA17.6F9 in combination with αCD28 clone 5B8 or the mitogen PMA with ionomycin produced better activation and expansion profiles, better transduction,

**Data availability statement:** All raw and meta data is available through the University of North Carolina. https://dataverse.unc.edu/dataverse/davisCART.

**Funding:** This work was supported by the National Institutes of Health through Grant Award R37-CA260223 and R33-CA281875 from the NCI, seed funding from the Comparative Medicine Institute and the Lineberger Cancer Center and by start-up funds provided by the University of North Carolina at Chapel Hill, North Carolina State University at Raleigh, and the Lineberger Cancer Center.

**Competing interests:** The authors declare the following financial interests/personal relationships which may be considered as potential competing interests: Yevgeny Brudno has patents related to CAR T cell production and therapy pending to NC State. Yevgeny Brudno is the scientific founder of Persistence Therapeutics, which seeks novel methods for CAR T cells therapy. None of the other authors have known competing financial interests or personal relationships that could have appeared to influence the work reported in this paper.

and more desirable T cell subsets that are more likely to improve patient outcomes in dogs suffering from round cell and solid tumors.

## Introduction

Animal models for human disease are instrumental in the development of novel therapeutics and treatment modalities. In particular, dogs are gaining popularity as a model for human neoplasia. Many of the neoplastic processes that occur spontaneously in the dog result from similar gene mutations, have similar courses of disease progression, similar prognoses, and respond to similar therapeutic regimens [1–3]. As such, exploration of novel immunotherapeutics in veterinary medicine using canine patients can offer insights into the treatment of round cell and solid tumors alike. Successfully troubleshooting hurdles to immunotherapy in dogs has the potential to bridge both cost and treatment failure gaps that limit immunotherapeutic advancement in humans.

T cell immunotherapies, especially chimeric antigen receptor (CAR) T cell therapy, are revolutionizing treatment of human hematological cancers, but their application in the veterinary space remains limited. Adoptive transfer of autologous canine T cells has been utilized experimentally in the treatment of B cell lymphoma [4] and osteosarcoma [5] with some improvement in patient outcomes. CAR T cells targeting canine CD20 were used experimentally at low doses for adjunct treatment of diffuse large B cell lymphoma and resulted in a decrease in target antigen expression [6]. These early studies toward canine CAR T cell treatments have been hampered by limited transduction attributable to suboptimal T cell stimulation. Not only is adequate stimulation of isolated autologous T cells necessary for activation and expansion *ex vivo* in order for patients to receive adequate therapeutic dosing, it is essential for high transduction rates for patients receiving genetically modified T cells.

Normal T cell stimulation occurs in lymphoid tissues where T cells encounter antigen presenting cells (APCs) that present peptides from endocytosed and digested foreign antigens. Primary stimulation is essential for T cell activation and occurs when the peptide binds to the T cell receptor (TCR) which is intimately associated with the transmembrane CD3 complex. Co-stimulatory signals, which boost CD3-mediated intracellular signaling to promote activation, are generated when B7 molecules on APCs bind CD28 expressed on the surface of naive T cells [7]. In the context of cancer immunity, CD28 stimulation is especially important as it induces anti-apoptotic signaling [8,9] and makes T cells more resilient within the tumor microenvironment where localized intratumoral acidification inhibits T cell function, proliferation and cytokine release [10–12].

Many strategies have been utilized to mimic this APC-T cell interaction to stimulate canine T cells *in vitro,* but there is room for improvement. Co-culture with γ-irradiated artificial APCs (aAPCs) engineered to stimulate canine T cells has contributed to excellent T cell activation [6] and expansion [13,14]. However, transduction with aAPC stimulation alone was relatively low, producing about 16% transduction [15]

with higher rates (17–37%) reported when cultured in the presence of the mitogen phytohemagglutinin (PHA), likely because of the additional stimulation [16].

Mitogen-based stimulation methods alone have shown potential in canine T cell transduction. PHA in the absence of aAPCs can generate retrovirally transduced T cells with highly variable results. Transduction rates of approximately 6% was reported by Cao [15], with Sakai reporting peak transduction of 67% [17] and 73% [13] in separate studies. Though these results are encouraging, transduced T cells from PHA stimulation have notoriously underperformed in human clinical trials, which may be true for veterinary patients as well [18]. While use of other mitogens like Concanavalin A (ConA) or phorbol myristate acetate (PMA) and ionomycin (PMA/I) have been described for canine T cell stimulation, their use has been limited. ConA has largely been relegated to use as a generic activation strategy, often for experimental positive controls [14,19,20]. PMA/I has been shown to reliably initiate canine T cell activation and proliferation [21], but this mitogen combination has been used primarily to induce cytokine release by T cells in culture [20,22–24]. While the availability and reliability of theses mitogens as T cell activators make them attractive candidates for canine T cell transduction, their use for this purpose has been limited to data from two canine donors, where ConA stimulation resulted in reasonable (28–49%) transduction and PMA/I resulted in low (10–26%) transduction [17]. Transduction using these mitogens shows promise, but their use is underexplored, justifying further analysis.

While mitogen-induced transduction should be investigated as it may have clinical benefit in veterinary medicine, antibody induced stimulation strategies have served as the most common method for human T cell immunotherapy [25–27]. As such, optimizing antibody-induced canine T cell stimulation is important for translational studies. Antibody-induced stimulation has been described for decades in human literature utilizing several strategies. Stimulatory anti-CD3 (αCD3) and anti-CD28 (αCD28) antibodies have been conjugated through covalent or high affinity bonding to a variety of bead types, including MACSibeads [28,29], Dynabeads [30–32], and biodegradable nanobeads [20]. Binding antibody to polystyrene plates through hydrophobic and electrostatic interactions has persisted as a method for T cell stimulation due in part to its ease of use [30,33], and advancements in surface treatment of polystyrene has improved predictability in antibody orientation [34] for optimal T cell engagement. Bead- [6,14,15,20] and plate-bound [13,21,35] anti-canine CD3 and anti-canine CD28 antibodies have been utilized with some success in canine T cells, but activation, expansion, and transduction have been highly variable depending on the antibody combination utilized and the strategy of antibody presentation.

Only two αCD3 antibody clones – CA17.2A12 (2A12) and CA17.6F9 (6F9) – and two αCD28 clones – 1C6 and 5B8 – have been described for use in canine T cell studies. 2A12 was first reported in 1993 by Moore and Rossitto [36] to aid in the diagnosis of T cell neoplasia. The origin of 6F9 is less well documented, but it was supplied by Moore's group and used by Graves et al. in 2011 [37], where it was used in combination with the newly developed αCD28 antibodies 5B8 or 1C6 to promote T cell expansion. In that context, expansion was enhanced by 5B8 co-stimulation, while it was inhibited by 1C6. As a result, 1C6 was initially characterized as antagonistic, but it has since been used by multiple groups to successfully stimulate canine T cells [17,21,35,38]. Despite decades of availability, few combinations of these antibodies have been thoroughly evaluated in the literature. As a result, an optimal strategy for antibody-induced canine T cell stimulation has not yet been defined.

The central aim of our work was to identify and optimize a stimulation protocol yielding robust canine T cell activation for optimized cell transduction and expansion towards benefiting canine patient T cell immunotherapy. As such, we sought to directly compare the efficacy of all possible combinations of stimulatory antibodies for canine CD3 and CD28 described in the literature. We determined the best strategy for canine T cell stimulation *ex vivo* by directly comparing bead- and plate-bound antibody strategies and the mitogens ConA and PMA/I. These approaches were studied in the context of T cell activation, expansion, and transduction as well as cell phenotypic changes. We concluded that plate-bound antibody was superior to bead-bound for T cell stimulation. Additionally, we concluded that PMA/I activation, followed closely by antibody-induced activation utilizing 5B8 αCD28 co-stimulation, produced optimal T cell transduction and expansion. 6F9

in combination with 58B showed improved expansion over 2A12 with 5B8 and reduced the proportion of unfavorable T cell subsets.

## Materials and methods

### Canine PBMC isolation and cryopreservation

Blood was collected by North Carolina State University's Lab Animal Resources veterinary team or donated by North American Veterinary Blood Bank in blood bags containing citrate phosphate dextrose adenine (CPDA-1) solution. 15 mL Lymphoprep (Accurate Chemical) was added to 50 mL conical tubes. Blood was diluted 1:1 with 1x Dulbecco's phosphate-buffered saline (DPBS) (Gibco). 25 mL of dilute blood was added to the Lymphoprep layer and centrifuged for 40 minutes at 400 x g with lowest acceleration setting and no brake. The peripheral blood mononuclear cell (PBMC) layer was transferred to clean 50 mL tubes and washed twice with DPBS. The PBMC pellet was placed on ice for 10 minutes prior to resuspension in cold freeze media containing 50% FBS, 40% complete canine T cell media, and 10% DMSO (Sigma). Cell suspension aliquots were frozen at −80°C before transferring to liquid nitrogen for long term storage. Fresh blood samples were not utilized in these experiments as direct comparisons between the same donors within and across experiments spanning several months were a major focus of this research.

### Canine T cell culture

Cryopreserved PBMCs were thawed and placed in flasks containing complete canine T cell media (cTCM) prepared with Advanced RMPI 1640 (Gibco) supplemented with 10 mmol/L HEPES (pH 7.3, ThermoScientific), 2 mmol/L L-Glutamine (Gibco), 100 U/mL Penicillin and 100 U/mL Streptomycin (Gibco), and 10% Fetal Bovine Serum (Cytvia, Hyclone Laboratories). Cells were left undisturbed overnight prior to stimulation for all assays. At the time of stimulation, PBMCs were resuspended in cTCM containing recombinant human IL-2 (PeproTech) at 200 U/mL. For experiments requiring a 14-day expansion period, PBMCs were transferred to cTCM free of stimulatory mitogen/antibody containing 500 U/mL IL-2, as supplementation of IL-2 at higher concentrations has shown to improve canine T cell expansion over 14 days [35]. For IL-2, unit measurement was determined by biological activity (according to PeproTech datasheet information), where the ED50 as determined by the dose-dependent stimulation of murine CTLL-2 cells is ≤ 0.1 ng/ml, corresponding to a specific activity of ≥ 1 x $10^7$ units/mg. Recombinant human IL-2 was selected over canine because it has demonstrated robust canine T cell expansion potential [16,35,39] and has resulted in superior transduction efficiency in canine T cells over recombinant canine IL-2 [15]. All culture was performed at 37°C, 5% $CO_2$, in a humidified incubator, with the exception of those experiments directly comparing the impact of 37°C versus 38.8°C on canine T cell transduction and expansion.

### Mitogen preparation and use for activation assay

Stock solutions of mitogens were prepared as follows and stored at −20°C until use: ConA (Sigma-Aldrich) at 5 mg/mL in DPBS, PMA (Sigma) at 1 mg/mL in DMSO (Sigma-Aldrich), and ionomycin calcium salt (Tocris) at 1 mg/mL in DMSO. For PMA/I, at the time of stimulation, working solutions were created by diluting the stock solutions in DPBS before adding to cells.

To determine optimal concentrations of mitogens for an activation assay, $10^6$ PBMCs from three healthy canine donors (S1 Table) were transferred to 24 well plates containing 2 mL IL-2 enriched (200 U/mL) cTCM with either ConA or PMA/I at varying concentrations. ConA concentrations of 0.5, 1, 2.5, or 5 μg/mL were selected based on previously described use [6,19,20,40]. Kato [39] demonstrated the negative impact on canine T cell expansion with ConA concentrations above 5 μg/mL, so this concentration was not exceeded. PMA/I concentrations of 2/0.5, 15/0.5, or 25 ng/mL/1 μg/mL were selected based on previously described use [17,20,41]. Mitogen-free cCTM was used as a negative control. Samples were placed in a humidified incubator at 37°C, 5% $CO_2$. After three days, CD25 expression was evaluated by flow cytometry to

determine activation frequency. Evaluating CD69 expression was considered, but anti-canine CD69 antibodies are not currently commercially available and reliable cross-reactive CD69-specific monoclonal antibodies used in other species have not been identified [21].

The mitogen dose impact on viability was also determined by flow cytometry using the gating strategy shown in S1 Fig. Viability, then, was measured as frequency of viable singlets and compared across mitogen concentrations.

## Antibody-bead conjugation and use for activation assay

Stimulatory anti-canine CD3 antibodies (clone CA17.2A12, Invitrogen; clone CA17.6F9, generously gifted by Peter Moore, UC Davis) and anti-canine CD28 antibodies (clone 1C6, eBioscience; clone 5B8, purified into PBS from hybridoma generously gifted by Brian Hayes, Fred Hutchinson Cancer Research Center), or non-stimulatory mouse IgG1 kappa Isotype control antibodies (clone P3.6.2.8.1, eBioscience, Invitrogen), were biotinylated using Mix-n-Stain Biotin Antibody Labeling Kit (Biotium) according to product instructions. To ensure adequate biotinylation of αCD3/αCD28 antibodies, PBMCs were labeled with either biotinylated or non-biotinylated antibody and then with secondary labeling using streptavidin-RPE (Serotec) or goat anti-mouse IgG1 cross adsorbed-PE (Invitrogen) respectively. Cells were then processed by flow cytometry and PE signaling from biotinylated and non-biotinylated antibody was virtually identical indicating successful antibody biotinylation.

Biotinylated antibodies were then conjugated to Anti-Biotin MACSiBead Particles (Miltenyi Biotec) according to manufacturer instructions at a concentration of 30 µg total biotinylated antibody per $1x10^8$ MACSiBead Particles. αCD3/αCD28 antibody beads were created by adding 15 µg αCD3 and 15 µg αCD28 to $1x10^8$ MACSiBead Particles. αCD3 antibody beads were created by adding 15 µg αCD3 and 15 µg isotype antibody to $1x10^8$ MACSiBead Particles. Isotype antibody was included to promote equidistant distribution of αCD3 on the bead surface of these beads and the αCD3/αCD28 antibody beads. Non-stimulatory isotype antibody beads were created by adding 30 µg isotype antibody to $1x10^8$ MACSi-Bead Particles. Successful conjugation was verified using flow cytometry. Because antibodies conjugated to beads were biotinylated, streptavidin-PE staining successfully identified conjugated beads, while unconjugated beads yielded no PE signaling (S2 Fig).

An activation assay was used to determine optimal concentration of bead-bound antibody for high activation efficiency for subsequent experiments. $10^6$ PBMCs from three healthy canine donors were transferred to 24 well plates in 2 mL IL-2 enriched media (200 U/mL). Antibody-conjugated beads were then added at varying doses so that for combination antibody beads, total antibody (αCD3 + αCD28) doses of 0.25, 0.5, 1, and 2 µg/well were used, which equated to cell:bead ratios of 1:0.825, 1:1.65, 1:3.3, and 1:6.6 respectively. For solitary antibody beads, αCD3 doses of 0.125, 0.25, 0.5, and 1 µg/well were used, which equated to cell:bead ratios of 1:0.825, 1:1.65, 1:3.3, and 1:6.6 respectively. ConA at 1 µg/mL was used as a positive control and isotype beads were used as a negative control at 2 µg/well. CD25 expression was evaluated after three days by flow cytometry.

## Antibody-coated plate preparation and use for activation assay

Anti-canine CD3 antibodies (clone CA17.2A12; clone CA17.6F9) and anti-canine CD28 antibodies (clone 1C6; clone 5B8-APC, eBioscience, Invitrogen) or IgG1 kappa Isotype control antibodies (clone P3.6.2.8.1) were utilized. Antibody was bound to nontreated polystyrene 24-well plates by adding αCD3 and αCD28 antibodies (1:1 ratio) to distilled water at the desired concentration. 500 µL of antibody solution was added to each well and the well plate was placed in a humidified incubator at 37°C, 5% CO2 for 4 hours. Plates were then removed and the antibody solution was aspirated from the wells. 2 mL cTCM containing 10% FBS was added to each well and incubated for 15–30 minutes at 37 °C in incubator as a blocking step [42] and to remove free antibody prior to addition of PBMCs.

An activation assay was used to determine optimal concentration of plate-bound antibody for high activation efficiency for subsequent experiments. Antibody solution was added to 24 well plates at varying doses, so that total antibody

(αCD3 + αCD28) dose was 0.25, 0.5, 1, or 2 μg/well. When αCD3 or αC28 antibody was evaluated alone, antibody solution was added to 24 well plates at 0.125 μg, 0.25 μg, 0.5 μg, or 1.0 μg/well. Non-stimulatory isotype IgG1 antibodies were used as a negative control at doses matching the highest total stimulatory antibody. ConA stimulation at 1 μg/mL was utilized as a positive control. PBMCs from three healthy canine donors were resuspended in IL-2 enriched media (200 U/mL). $10^6$ PBMCs in 2 mL of media were added to each well containing bound antibody and returned to the incubator. CD25 expression was evaluated by flow cytometry after three days. The antibody dose impact on viability was evaluated as previous (S1 Fig) and viability was compared across antibody doses.

## 4-day proliferation assay with CFSE analysis

PBMCs from three healthy canine donors were labeled with CellTrace CFSE Cell Proliferation Kit (Invitrogen) according to manufacturer instructions. PBMCs were then resuspended in IL-2 enriched media (200 U/mL) and stimulated with optimal concentrations of ConA, PMA/I, plate- or bead-bound αCD3 antibodies alone or in combination with αCD28 antibodies. Plate- and bead-bound isotype antibodies were used as negative controls. Cell counts were performed using CellDrop (DeNovix) at 24, 48, and 96 hours during the stimulation period. At 96 hours, cells were removed from stimulation and the number of cell divisions was determined by evaluating CFSE intensity by flow cytometry. For samples containing antibody conjugated beads, MACSibeads were removed with Dynal MPC-S (Magnetic Particle Concentrator) (Invitrogen) prior to flow cytometry.

## Transduction assay

PBMCs from three healthy canine donors were stimulated with optimal concentrations of ConA, PMA/I, plate- or bead-bound αCD3 antibodies alone or in combination with αCD28 antibodies in IL-2 enriched (200 U/mL) media for three days. Plate- or bead-bound isotype antibody, and non-transduced stimulated PBMCs were used as negative controls. The day prior to transduction, 7 μL RetroNectin (TaKaRa) in 1 mL DPBS was added to each well of nontreated 24 well plates and incubated at 4°C overnight. The following day, RetroNectin was aspirated and plates were blocked with media containing 10% FBS prior to the addition of GFP-encoded GALV-pseudotyped gamma retrovirus (generously gifted by Gianpietro Dotti, University of North Carolina) at MOI 13. Plates were centrifuged at 2000xg for 90 minutes, after which viral supernatant was removed. MACSibeads were removed where necessary and PBMCs were washed and resuspended in fresh IL-2 enriched media. Cell suspension added to the wells of the RetroNectin/virus coated plates at 7 x $10^5$ cells/2 mL. Plates were centrifuged at 1000xg for 10 minutes and placed in an incubator. All samples were limited to a single round of transduction. After three days, GFP expression was evaluated by flow cytometry to determine transduction efficiency.

## 14-day proliferation assay with T cell phenotype analysis

PBMCs from five canine donors were stimulated with ConA, PMA/I, or plate-bound CA17.2A12 or CA17.6F9 in combination with 5B8 in IL-2 enriched (200 U/mL) media for three days. PBMCs were then washed and resuspended in fresh high concentration IL-2 enriched (500 U/mL) media. $10^6$ cells from each donor were transduced once as described with GFP gamma retrovirus (MOI 10). Cells were expanded in culture free of stimulatory mitogen or antibody for 14 days. IL-2 enriched media was replaced every three days. Cell counts were taken on day 0 (day of transduction), 3, 7, 10, and 14. Cell viability was assessed using Trypan Blue solution (Gibco) and CellDrop. CD4/CD8 T cell and regulatory T cell ($T_{reg}$) presence were evaluated by flow cytometry 24 hours after thaw (naive), 7 days post-transduction, and 14 days post-transduction. $T_{regs}$ were classified as CD4+CD25+Foxp3+ cells [43–45]. Intracellular GFP expression and median fluorescence intensity were evaluated 4 days after transduction to confirm success and extent of transduction. Finally, GFP expression was evaluated at 14 days and the phenotype of transduced cells was compared to the general T cell population.

## Flow cytometry

Cell suspension was collected and washed once with DPBS and centrifugation at 400xg for 5 minutes. To evaluate viability, LIVE/DEAD Near IR Fixable Dead Cell Stain Kit (Invitrogen) or Live-or-Dye 405/452 Fixable Viability Staining Kit (Biotium) was prepared at 1:1000 dilution in DPBS. Cells were resuspended in diluted viability dye and incubated in the dark at 4°C for 30 minutes. Cells were washed and resuspended in antibody solution containing rat anti-canine CD5 (clone YKIX322.3)-PerCP-eFluor 710 (Invitrogen, eBioscience), mouse anti-canine CD25 (clone P4A10)-Super Bright 600 or PE (Invitrogen, eBioscience), and/or rat anti-canine CD4 (clone YKIX302.9)-APC (Invitrogen, eBioscience). In the event of primary antibody with no conjugated fluorophore, mouse anti-canine CD3e (clone CA17.2A12, Invitrogen) labeled cells were incubated in the dark at 4°C for 30 minutes. Cells were washed followed by the addition of antibody solution containing goat anti-mouse IgG1 cross-adsorbed secondary antibody-PE (Invitrogen) and stained cells were incubated in the dark at 4°C for 30 minutes. After indirect labeling, cells were washed twice in stain buffer followed by the addition of solution containing the following conjugated antibodies: rat anti-canine CD8a (clone YCATE55.9)-Super Bright 436 (Invitrogen, eBioscience) and rat anti-canine CD4 (clone YKIX302.9)-APC (Invitrogen, eBioscience). Stained cells were incubated in the dark at 4°C for 30 minutes. All antibody solutions to label cell surface markers were prepared in Fetal Bovine Serum Stain Buffer (BD) at the appropriate antibody dilution based on titration studies using flow cytometry. With the exception of GFP expressing cells or cells undergoing fixation/permeabilization for Foxp3 intracellular labeling, cells were fixed using 250 mL Cytofix Fixation Buffer (BD). Cells were then washed once and resuspended in stain buffer for flow cytometry. For Foxp3 intracellular labeling, viability and cell surface marker staining were first performed followed by fixation and permeabilization using Foxp3/Transcription Factor Staining Buffer Set (Invitrogen, eBioscience). Rat anti-mouse Foxp3 (clone FJK-16s), eFluor™ 450 (Invitrogen, eBioscience) was diluted in permeabilization buffer and cells were incubated for 30 minutes in the dark at room temperature. Clone FJK-16s has demonstrated cross reactivity with canine Foxp3 [22]. Cells were then washed and resuspended in the desired amount of stain buffer for flow cytometry.

## Statistical analysis

Flow cytometry data was analyzed using FCS Express 6 Flow Research Edition software. Statistical analysis was performed using Graph Pad Prism 9 (10.1.2) software. Pairwise differences between groups were analyzed with Ordinary One-Way analysis of variance (ANOVA) or two-tailed t-test where appropriate. Multiple comparison statistical analysis was performed by Two-way ANOVA. Differences were deemed significant if $p < 0.05$.

## Figures

All figures were created using Graph Pad Prism 9 (10.1.2), FCS Express 6 Flow Research Edition, and BioRender.

## Ethics statement

Peripheral whole blood was either shipped by North American Veterinary Blood Bank (donated by privately owned dogs) or was collected by North Carolina State University's Lab Animal Resources veterinary team. All procedures involving animals were reviewed and approved by the Institutional Animal Care and Use Committee (IACUC) at North Carolina State University College of Veterinary Medicine (Protocol Number: 23-236). Blood samples were collected from dogs in accordance with approved institutional guidelines for animal care and use. All efforts were made to minimize stress and discomfort during sample collection, and appropriate handling techniques were employed throughout the study. No excessive or prolonged distress or pain was experienced by donors that would warrant sedation, anesthesia, or analgesics.

## Results

### Optimal concentrations of ConA and PMA/I are limited by impact on viability

T cell activation is essential to induce expansion and to promote effector function *in vivo* for clinical efficacy. To determine optimal concentrations for mitogen-mediated T cell activation, PBMCs from three canine donors were cultured with

different concentrations of ConA and PMA/I. After three days CD5$^+$ T cells were evaluated for the upregulation of the IL-2 receptor subunit CD25 as a marker for activation. It is well established that mitogens are potent T cell activators and that overstimulation can drive apoptosis through activation-induced cell death (AICD) [20,46,47]. We therefore determined optimal mitogen concentration as that which achieved the highest activation frequency without an unacceptable loss in viability. Using flow cytometry (S1 Fig), viability at each dose increase was compared to the lowest dose. Any dose increase resulting in a 10% or greater drop in viability was disqualifying.

T cells responded favorably to ConA with little variability in donor response at all concentrations (Fig 1A). Though the highest concentration of ConA produced the most robust T cell activation, doses of 2.5 μg/mL and above resulted in unacceptable loss in cell viability (Fig 1B). Therefore, a 1 μg/mL ConA concentration was determined to be optimal and used for subsequent experiments.

PMA/I stimulation showed a roughly linear relationship between T cell activation and mitogen dose. T cells responded well to stimulation at all concentrations. Mild variability in donor response observed at lower concentrations was completely resolved at the highest dose (Fig 1C). Because no unacceptable loss in cell viability was seen at the highest PMA/I concentration evaluated (Fig 1D), 25 ng/mL + 1 μg/mL (PMA + I) was selected for use in subsequent experiments.

## Bead-bound antibody with 5B8 co-stimulation is a superior inducer of canine T cell activation

Antibody-conjugated beads have demonstrated efficacy for T cell activation [20]. To determine optimal dosing for antibody-conjugated beads, we evaluated all published α-canine CD3 and α-canine CD28 antibodies. MACSiBead Particles were loaded with αCD3 clones CA17.2A12 (2A12) or CA17.6F9 (6F9) alone for primary stimulation or in combination with αCD28 clones 1C6 or 5B8 for co-stimulation. Beads were added to PBMCs from three canine donors at different doses and CD25 expression was evaluated after three days (Fig 2A).

Co-stimulation with 5B8 resulted in superior T cell activation over 1C6 and behaved similarly with both 2A12 and 6F9. Good activation was achieved with 2A12 + 5B8 and 6F9 + 5B8 beads at the highest doses of 2 μg/well, although theses activations were slightly lower than those seen with mitogens. Overall, 5B8 co-stimulation resulted in linear increase in activation that correlated well with dose increase. Of note, with 2A12 + 5B8 stimulation, Donor #2 response plateaued at 1 μg/well, which lead to higher donor variability at the highest dose. This was not seen with 6F9 + 5B8, where minimal donor variability was observed at the highest dose (Fig 2B and 2C).

Co-stimulation with 1C6 on beads showed different patterns when combined with 2A12 and 6F9 (Fig 2D and 2E). Efficacy of 2A12 + 1C6 was poor initially, but increased to 75.4% activation at 2 μg/well. This approached, but fell short of activation seen with 5B8 co-stimulation. T cells responded poorly to 6F9 + 1C6 bead stimulation with activation reaching only 36.7% at the highest dose.

As the highest dose resulted in the best activation for all αCD3 + CD28 beads, 2 μg/well was deemed the optimal dose and was used in subsequent experiments.

Although primary αCD3 stimulation alone has been reported to induce T cell activation and proliferation [37,48], very little activation was observed with αCD3-bound beads. Stimulation with 6F9 beads alone did not promote T cell activation even at the highest dose (Fig 2F). Some (~32%) T cell activation was achieved with 2A12 beads alone at the highest dose of 1 μg/well (Fig 2G), but this was well below that achieved with 5B8 or 1C6 co-stimulation.

Taken together, these data demonstrate the importance of co-stimulation for bead-bound antibody-induced T cell activation. The clear increase in activation using co-stimulation with the 5B8 clone will be important for future T cell treatments in canine patients and suggests it should be used in favor of the currently commonly used 1C6 clone [20,38].

## Plate-bound αCD3 in combination with 5B8 co-stimulatory antibody outperforms beads for canine T cell activation

Plate-bound antibodies are commonly used to activate T cells and minimize the expense and labor associated with bead conjugation [21]. To determine the optimal dose of stimulatory antibodies on coated plates for antibody-induced T cell

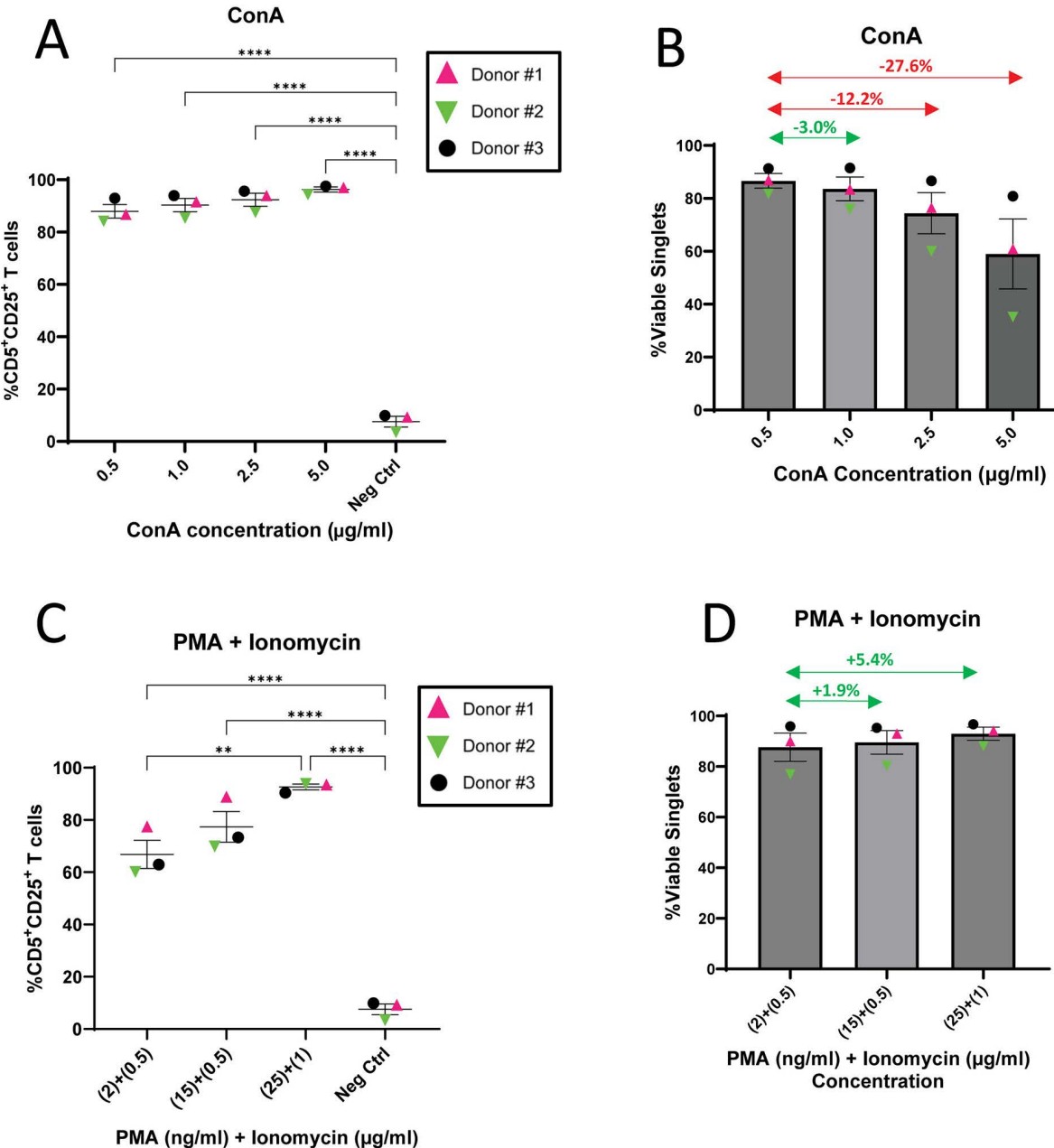

**Fig 1. Optimal concentrations of ConA and PMA/I are limited by impact on viability. (A–D)** Mitogens were used to activate PBMCs (n = 3) at different concentrations to determine optimal dose. CD5+ T cell activation was determined by cell surface expression of CD25 and viability was evaluated as an indicator for activation-induced cell death. **(A)** Frequency of activated T cells after ConA stimulation. **(B)** Impact of ConA dose increase of on cell viability. Arrows indicate the difference in cell mortality at each dose compared to the lowest dose. Green arrows indicate less than 10% loss in viability, which was deemed acceptable. Red arrows indicate a greater than 10% loss in viability which was disqualifying. **(C)** Frequency of activated T cells after PMA/I stimulation. **(D)** Impact of PMA/I dose increase of on cell viability. For **(A, C)** horizontal lines indicate mean values, and error bars represent standard error of the mean. Pairwise statistical analysis was performed by One-way ANOVA with Tukey's multiple means comparison; ** p < 0.01, **** p < 0.0001. For **(B, D)**, viability was defined as the frequency of viable singlets as determined by flow cytometry. Horizontal lines indicate mean values, and error bars represent standard error of the mean.

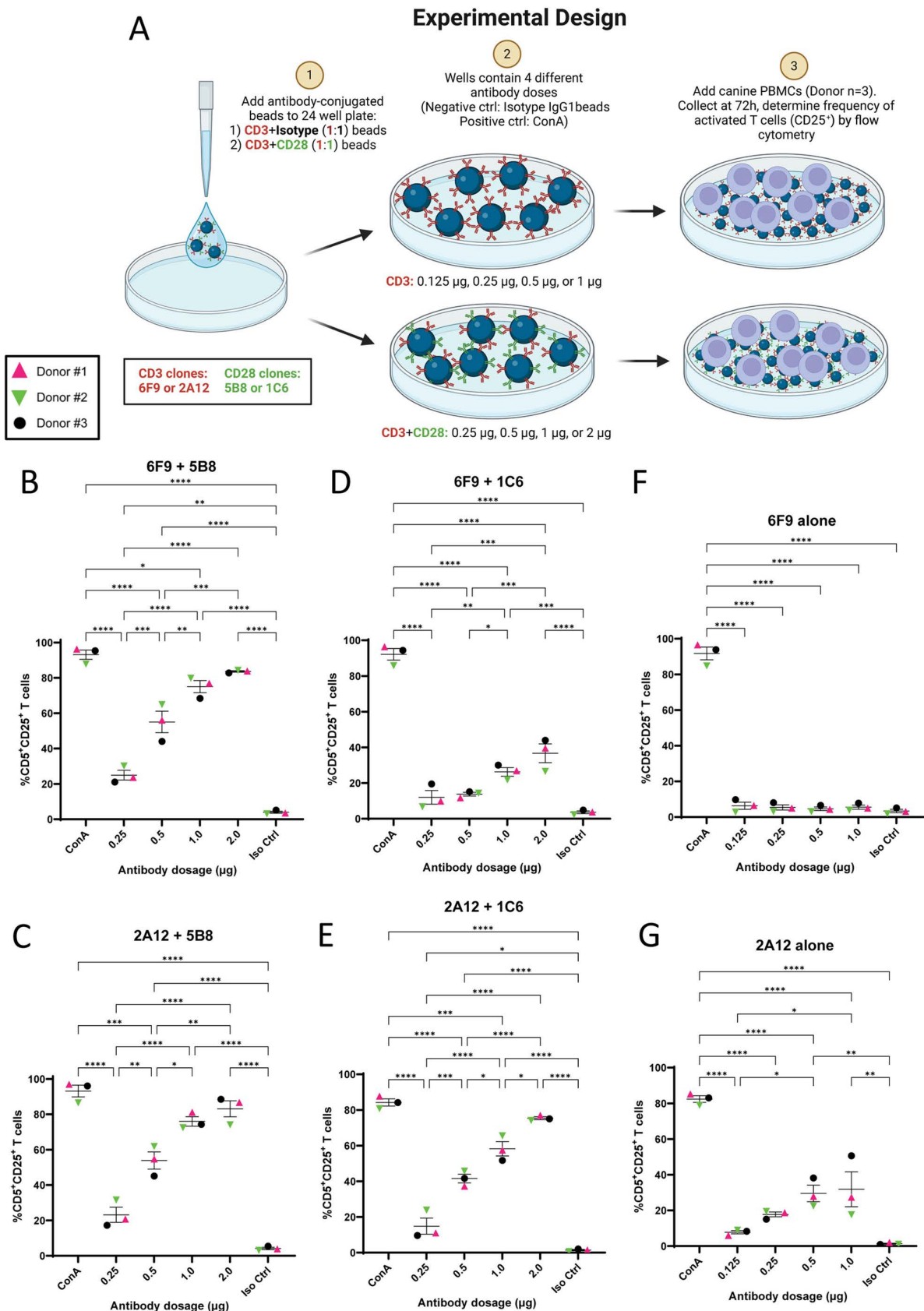

**Fig 2. Bead-bound antibody utilizing 5B8 co-stimulation demonstrates superior T cell activation. (A)** Schematic representation of experimental design. Bead-bound stimulatory antibodies (6F9=CA17.6F9; 2A12=CA17.2A12) were used to activate PBMCs (n = 3) and frequency of activated $CD5^+$ T cells was determined by cell surface expression of CD25. **(B-G)** Different antibody doses were utilized to determine optimal dose for superior T cell activation. The stimulatory effects of four different αCD3 + αCD28 bead-bound antibody combinations (1:1 ratio) and two different αCD3 + isotype bead-bound antibodies (1:1 ratio) were evaluated: **(B)** 6F9 + 5B8 beads, **(C)** 2A12 + 5B8 beads, **(D)** 6F9 + 1C6 beads, **(E)** 2A12 + 1C6 beads, **(F)** 6F9 beads, and **(G)** 2A12 beads. Horizontal lines indicate mean values, and error bars represent standard error of the mean. Pairwise statistical analysis was performed by One-way ANOVA with Tukey's multiple means comparison; * p < 0.05, ** p < 0.01, *** p < 0.001, **** p < 0.0001.

activation, αCD3 antibodies (2A12 or 6F9) were added to polystyrene 24-well plates alone or in combination with αCD28 antibodies (1C6 or 5B8) at different doses. Canine PBMCs from three donors were added to the plates and CD25 expression was evaluated after three days of stimulation (Fig 3A).

Plate-bound αCD3-stimulated PBMCs responded quite favorably to co-stimulation with 5B8. Interestingly, a lag in responsiveness was seen in Donor #2 at lower doses, which was not seen with bead-induced stimulation. This donor variability was reduced as antibody dose increased and was completely resolved at 1 µg/well and above. With 5B8 co-stimulation, T cell activation was most robust at the highest antibody concentration of 2 µg/well for both 2A12 and 6F9, with T cell activation reaching 97.3% and 97.8% respectively, both of which exceeded ConA-induced activation (Fig 3B and 3C).

Co-stimulation with plate-bound 1C6 yielded much less impressive results. Response to stimulation was indistinguishable from negative controls at the lowest doses with both 2A12 and 6F9. Very little increase in activation was seen with 6F9 as the dose increased and a peak T cell activation of only 17.9% was achieved (Fig 3D). A better response to dose increase was noted with 2A12, where peak T cell activation of 50.3% was achieved at 2 µg/well, though this underperformed bead-bound antibodies of the same combination. Donor #2 again responded poorly to 2A12 + 1C6 stimulation compared to other donors. This lag in responsiveness was far more exaggerated than with 5B8 co-stimulation, and did not resolve at higher doses (Fig 3E).

In all instances, the highest dose of plate-bound αCD3 + CD28 antibody resulted in the most robust activation, so 2 µg/well was deemed the optimal dose and used in subsequent experiments.

Similar to bead-bound antibodies, primary stimulation with plate-bound 6F9 alone did not induce T cell activation even at the highest dose (Fig 3F). In contrast, primary stimulation with plate-bound 2A12 alone was a dramatically more effective inducer of T cell activation than when it was bound to beads. In fact, Donor #1 and #3 response to 2A12 alone was comparable to ConA, though Donor #2 again was relatively a poor responder (Fig 3G). Paradoxically, with plate-bound antibodies, addition of 1C6 co-stimulation to 2A12 actually reduced activation as compared to 2A12 alone. This negative impact of 1C6 on activation with plate-bound 2A12 was reproducible and verified with a repeated experiment (S3A and S3B Fig). This is consistent with previous reports that found plate-bound 1C6 to be antagonistic when used with much higher doses of 6F9 than were evaluated here [37].

Taken together, this shows that plate-bound antibody with 5B8 co-stimulation resulted in extremely efficient activation that was better than bead-bound antibodies of the same combinations. As with beads, 5B8 outperformed 1C6 in the plate-bound format. Since early reports suggest both clones bind CD28 with similar efficiency [37], the difference in activity is likely due to variations in the binding site or binding affinity, resulting in enhanced intracellular signaling with 5B8.

In contrast to beads, where 1C6 co-stimulation enhanced activation, plate-bound 1C6 failed to improve activation over αCD3 antibodies alone, and in fact appeared to be inhibitory with 2A12. While the contradictory behavior of 1C6 antibody was perplexing, we propose that differences in antibody-substrate interaction and the subsequent presentation to T cells provide the most plausible explanation. High affinity antibody binding to MACSiBeads results in a substrate with predictable antibody orientation, high surface density, spherical topology, and dynamic mobility. Conversely, passive adsorption to polystyrene plates results in variable antibody orientation [34] and comparatively lower surface density [49] on a flat and rigid substrate. These contrasting conditions likely alter crosslinking behavior, create differential steric hindrance effects, and modify T cell receptor engagement, collectively modulating 1C6 downstream intracellular signaling.

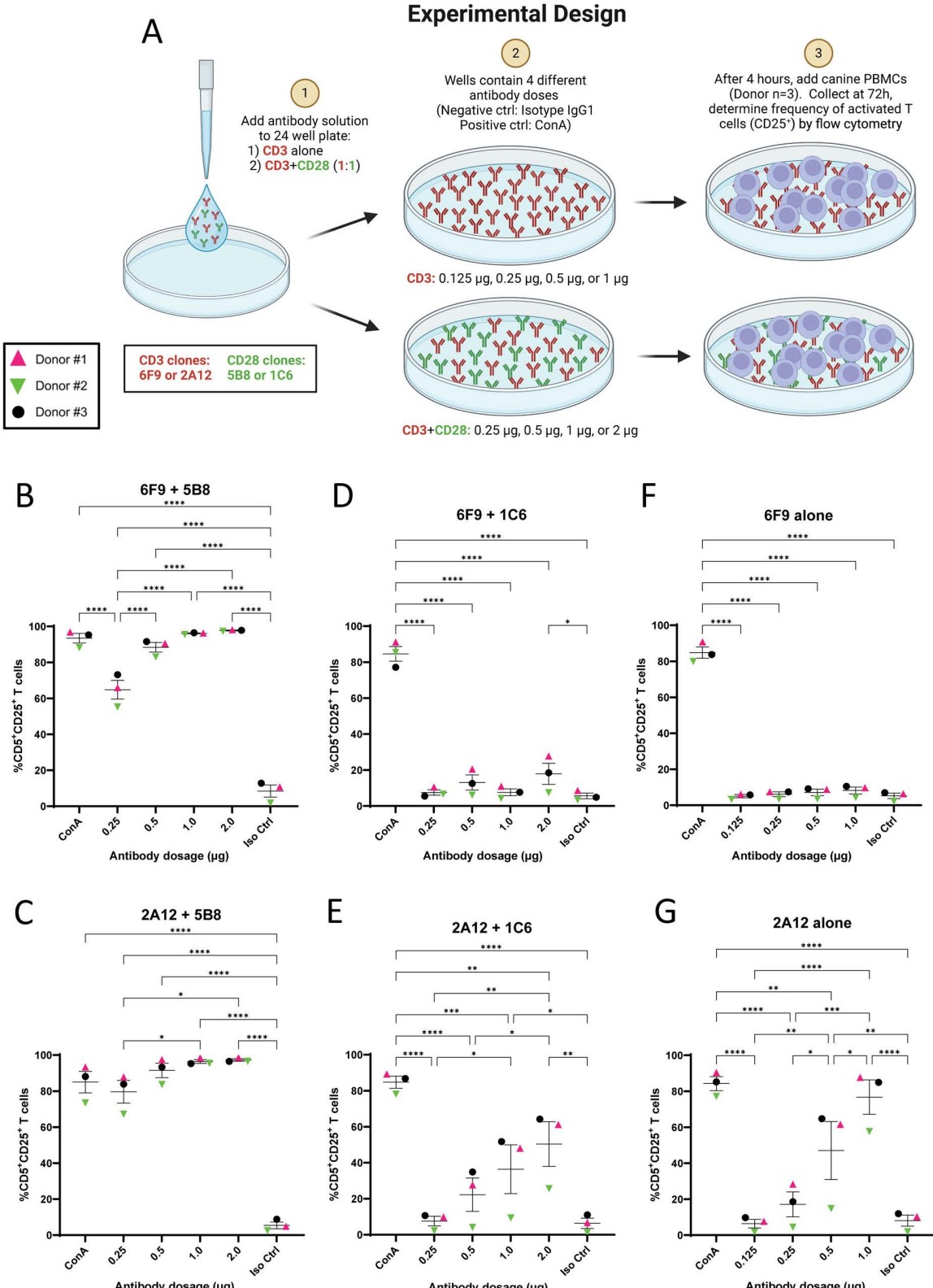

**Fig 3. 5B8 co-stimulation demonstrates superior T cell activation for plate-bound antibody. (A)** Schematic representation of experimental design. Plate-bound stimulatory antibodies (6F9=CA17.6F9; 2A12=CA17.2A12) were used to activate PBMCs (n = 3) and frequency of activated CD5⁺ T cells

was determined by cell surface expression of CD25. **(B-G)** Different antibody doses were utilized to determine optimal dose for superior T cell activation. The stimulatory effects of four plate-bound αCD3 + αCD28 antibody combinations (1:1 ratio) and two solitary plate-bound αCD3 antibodies were evaluated: **(B)** 6F9 + 5B8, **(C)** 2A12 + 5B8, **(D)** 6F9 + 1C6, **(E)** 2A12 + 1C6, **(F)** 6F9 alone, and **(G)** 2A12 alone. Horizontal lines indicate mean values, and error bars represent standard error of the mean. Pairwise statistical analysis was performed by One-way ANOVA with Tukey's multiple means comparison; * $p < 0.05$, ** $p < 0.01$, *** $p < 0.001$, **** $p < 0.0001$.

In light of the co-stimulatory function of 5B8 on plates, we investigated whether αCD28 co-stimulation alone could activate canine T cells. As expected, co-stimulation with 5B8 or 1C6 αCD28 antibodies alone was insufficient to induce T cell activation even at the highest dose (S4A and S4B Fig), underscoring the requirement for primary stimulation.

Having established that plates with 5B8 co-stimulation confer more robust activation than beads, we also confirmed that this robust activation did not induce AICD. The impact of dose increase on cell viability was explored and even at the highest doses evaluated for antibody stimulation, there was no apparent loss in cell viability for either 2A12 or 6F9 in combination with 5B8 on plates (S5A and S5B Fig).

### Direct comparison of the optimal doses of mitogen and stimulatory antibodies on canine T cell activation

Fig 4 summarizes activation achieved at optimal doses of all mitogens, plate-, and bead-bound antibody combinations. As expected, mitogen stimulation with ConA and PMA/I produced excellent T cell activation. Generally speaking, in the case of αCD3- and αCD28-induced T cell activation, co-stimulation with 5B8 was superior to 1C6 and greatly improved the activation with both plate-bound and bead-bound strategies. Plate-bound antibodies outperformed bead-bound antibodies with 5B8 co-stimulation, though this did not reach statistical significance. Interestingly, activation with the less effective 1C6 co-stimulatory antibody worked much more effectively with a bead-bound antibody strategy over plate-bound antibodies (S6A, S6B and S7 Figs). Taken together, these data suggest that, among antibody strategies, plate-bound 2A12 or 6F9 with 5B8 co-stimulation improves canine T cell activation, reaching statistical significance in most cases.

### Exogenous IL-2 alone is insufficient to induce canine T cell activation

Experiments on human T cells have demonstrated that supplementation with high doses of exogenous IL-2 alone is capable of inducing activation *in vitro* [50]. To better understand how IL-2 affects canine T cell activation, PBMCs from three donors were cultured without stimulation in media containing recombinant human IL-2 at 200 U/mL, 500 U/mL, or 2300 U/mL. Additionally, PBMCs were cultured with plate-bound 6F9 + 5B8 with or without IL-2 supplementation. Media without stimulatory antibody or IL-2 supplementation was used as a negative control. CD25 expression was evaluated at day 3 and 5.

Unlike the effects reported in human T cells, IL-2 alone was insufficient to activate canine T cells even at the highest doses evaluated. Antibody stimulation in the absence of exogenous IL-2 was sufficient to activate T cells after three days, although a slight drop in CD25 expression was noted after five days without the addition of exogenous IL-2, suggesting IL-2 plays a role in maintaining a state of activation (S8 Fig).

### Plate-bound 6F9 + 5B8 induces superior short-term canine T cell expansion

Expansion of T cells is critical for a successful immunotherapeutic response. To evaluate how differing stimulation strategies impact expansion behavior, PBMCs from the same three canine donors as previous were labeled with CFSE and stimulated for four days with optimal doses of mitogens, plate-, or bead-bound αCD3 antibody alone or in combination with αCD28. Cell counts were performed during the stimulation period and the number of cell divisions was determined after 96 hours by evaluating CFSE intensity by flow cytometry.

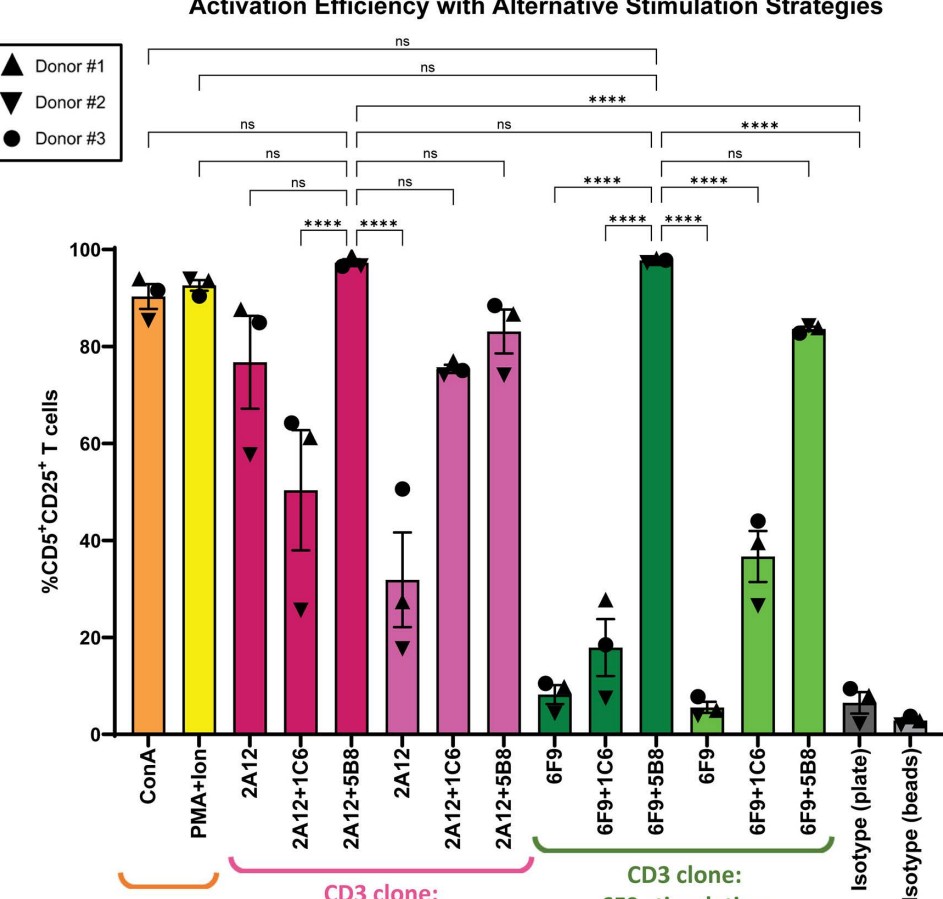

**Fig 4. Mitogen and plate-bound antibody with 5B8 co-stimulation outperform other T cell activation strategies.** At optimal doses of mitogens, plate-, and bead-bound antibodies (6F9=CA17.6F9; 2A12=CA17.2A12), this is a summary of all activation assays performed and directly compares the impact of these T cell stimulation strategies on canine T cell activation after three days of stimulation (n = 3). Because ConA and isotype antibodies were used as positive and negative controls respectively for the activation assays utilizing antibody stimulation, ConA and isotype values for Donors #1-3 represent the average activation efficiency for each donor. Horizontal lines indicate mean values, and error bars represent standard error of the mean. Pairwise statistical analysis was performed by One-way ANOVA with Tukey's multiple means comparison; **** p < 0.0001, ns not significant.

5B8 co-stimulation was important for optimal antibody-mediated T cell expansion with both plate- and bead-bound strategies. Though beads utilizing 5B8 co-stimulation provided better expansion than other bead-bound antibody strategies, they were much less effective than plate-bound antibodies of the same combinations (Fig 5A–5E). In fact, plate-bound antibodies using 5B8 co-stimulation resulted in a more robust proliferation profile than all other strategies, including mitogens, which demonstrated equivalent efficacy for T cell activation. 1C6 co-stimulation in general was least effective in inducing T cell proliferation, and seemed to impair proliferation in the case of plate-bound 2A12 + 1C6 compared to 2A12 alone, consistent with the activation assay data (Fig 5A and 5B). Donor variability in responsiveness to stimulation was noted, where Donors #1 and #2 responded similarly while T cells from Donor #3 were much more apt to expand, regardless of stimulation strategy (Fig 5B).

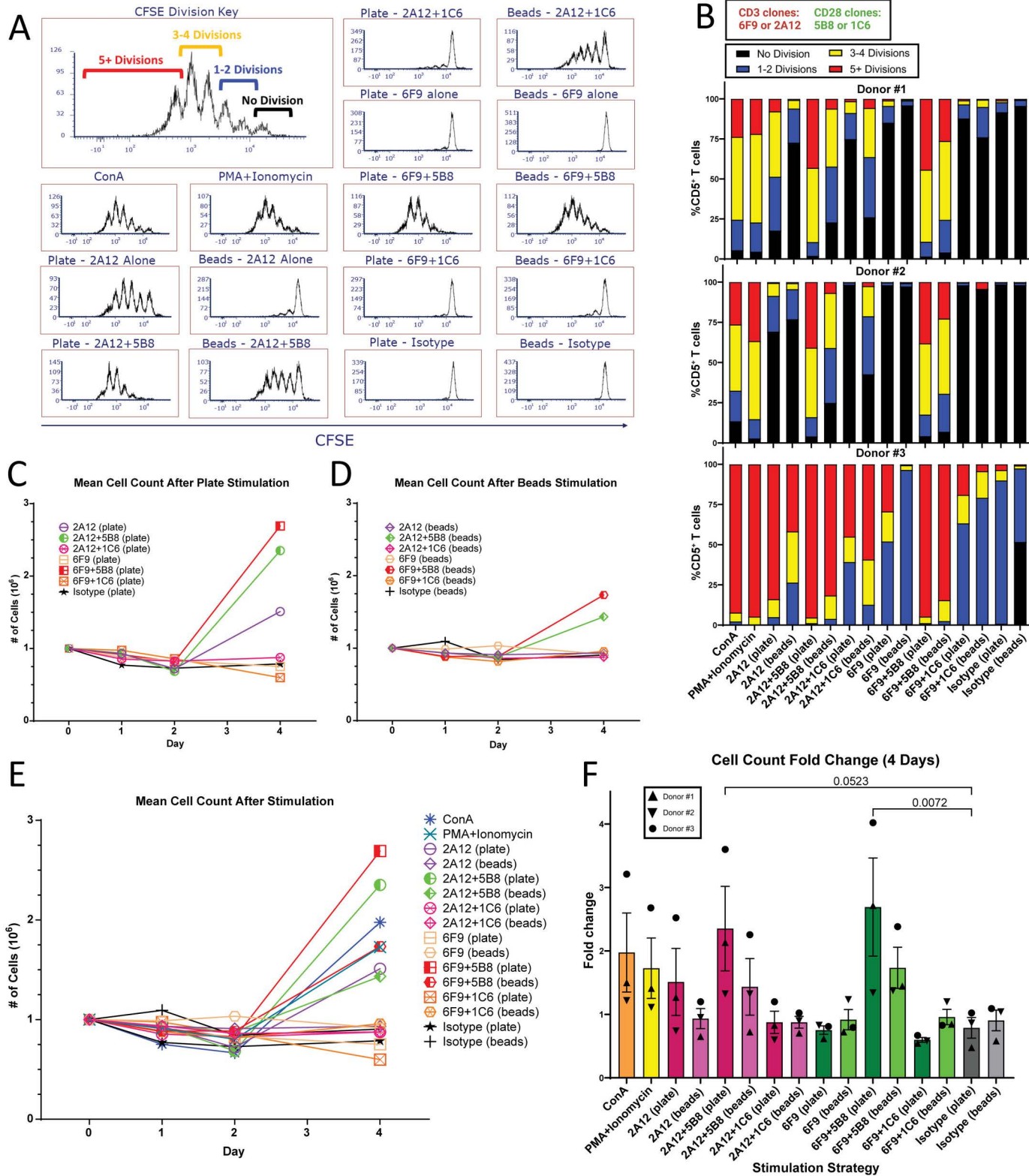

**Fig 5. Plate-bound antibody utilizing 5B8 co-stimulation results in a superior short term T cell expansion profile. (A-B)** CFSE proliferation assay (n = 3) over four days using mitogen, plate-, or bead-bound antibody (6F9=CA17.6F9; 2A12=CA17.2A12). CFSE intensity in CD5⁺ T cells was evaluated

on day 4. **(A)** Representative histogram (Donor #1) of proliferation profiles in CD5$^+$ T cells after utilizing different stimulation methods. **(B)** Graphical representation of CFSE proliferation assay illustrating the number of cellular divisions over four days in CD5$^+$ T cells for individual donors. **(C-E)** Degree of proliferation in PBMCs from stimulation was determined by cell counts over four days. All experimental samples started with 10$^6$ PBMCs on day 0. Each symbol represents the mean cell count on the corresponding day. **(C)** is an evaluation of antibody-coated plate stimulation only, **(D)** is an evaluation of antibody-conjugated bead stimulation only, and **(E)** evaluates all stimulation strategies concurrently. **(F)** Evaluates the fold-change between day 0 and day 4 in all donors. Horizontal lines indicate mean values, and error bars represent standard error of the mean. Pairwise statistical analysis was performed by One-way ANOVA with Tukey's multiple means comparison; $p < 0.05$ is considered significant.

Consistent with the previous experiment showing that IL-2 alone could not activate canine T cells, IL-2 alone also failed to drive unstimulated T cell expansion. Diminished expansion was also seen with stimulation strategies that resulted in poor activation. This suggests that in canine T cells, the high affinity IL-2 receptor (IL-2R) resulting from CD25 upregulation is crucial for robust proliferation in response to exogenous IL-2 [35,39].

Fig 5F summarizes fold expansion of T cells over four days using the different stimulation strategies. Only plate-bound 6F9 + 5B8 reached statistical significance over negative controls, though plate-bound 2A12 + 5B8 approached statistical significance. Of note, with the degree of T cell proliferation seen with superior stimulation strategies in the CFSE analysis (Fig 5B), we did not see a corresponding increase in fold-change expected in the total PBMC cell count (Fig 5F). This discrepancy was most likely attributable to the cell heterogeneity in PBMCs. The proportion of T cells, which would be the expanding population, would be variable in the starting PBMC population and was not determined.

Taken together, these data show that plate-bound αCD3 antibody with 5B8 co-stimulation drives better T cell expansion than all other strategies.

## Stimulation with PMA/I and plate-bound αCD3 with 5B8 co-stimulation results in superior T cell transduction

For cancer immunotherapy utilizing genetically modified T cells, high transduction efficiency enables optimal tumor-specific clinical responses. To determine the impact of different stimulation strategies on the transduction of canine T cells, PBMCs from the same three canine donors as previous were stimulated with optimal doses of mitogens, plate-, or bead-bound αCD3 antibody alone or in combination with αCD28. After three days, based on previous literature [13,15,17], PBMCs were transduced once with GFP-encoding gamma retrovirus and intracellular GFP expression was evaluated after three days.

Activated, proliferating T cells are far more likely to be transduced than naive T cell populations that have a low rate of proliferation [51]. In keeping with our proliferation data, it was not surprising that T cells from Donor #3 had higher transduction than T cells from other donors, regardless of stimulation strategy (Figs 5B and 6). PMA/I stimulation resulted in the most robust transduction with mean transduction reaching 72.6% (range 56.1–84.7%). Despite lower capacity for proliferation, this strategy surpassed both plate-bound 2A12 + 5B8 and 6F9 + 5B8, though this did not reach statistical significance. Transduction from PMA/I stimulation was significantly higher than all other stimulation strategies, including ConA which had a similar T cell expansion profile, but yielded only 41.3% transduction (range 30.1–56.7%) (Fig 6).

Stimulation with plate-bound 2A12 + 5B8 and 6F9 + 5B8 resulted in similar transduction at 59.3% (range 51.3–72.5%) and 59.8% (range 48.0–76.6%) respectively. Plate-bound antibodies with 5B8 co-stimulation resulted in significantly better transduction than all other plate or bead-bound antibody stimulation strategies (S9 Fig). This included bead-bound antibodies utilizing 5B8 co-stimulation, which yielded more than 2-fold lower transduction than plates of the same combination. Plate-bound antibodies with 5B8 co-stimulation trended toward higher transduction than ConA, but failed to reach statistical significance (Fig 6).

Given that PMA/I, plate-bound 6F9 or 2A12 with 5B8, and ConA were better than all other strategies for T cell activation, expansion, and transduction, only these strategies were utilized in subsequent experiments.

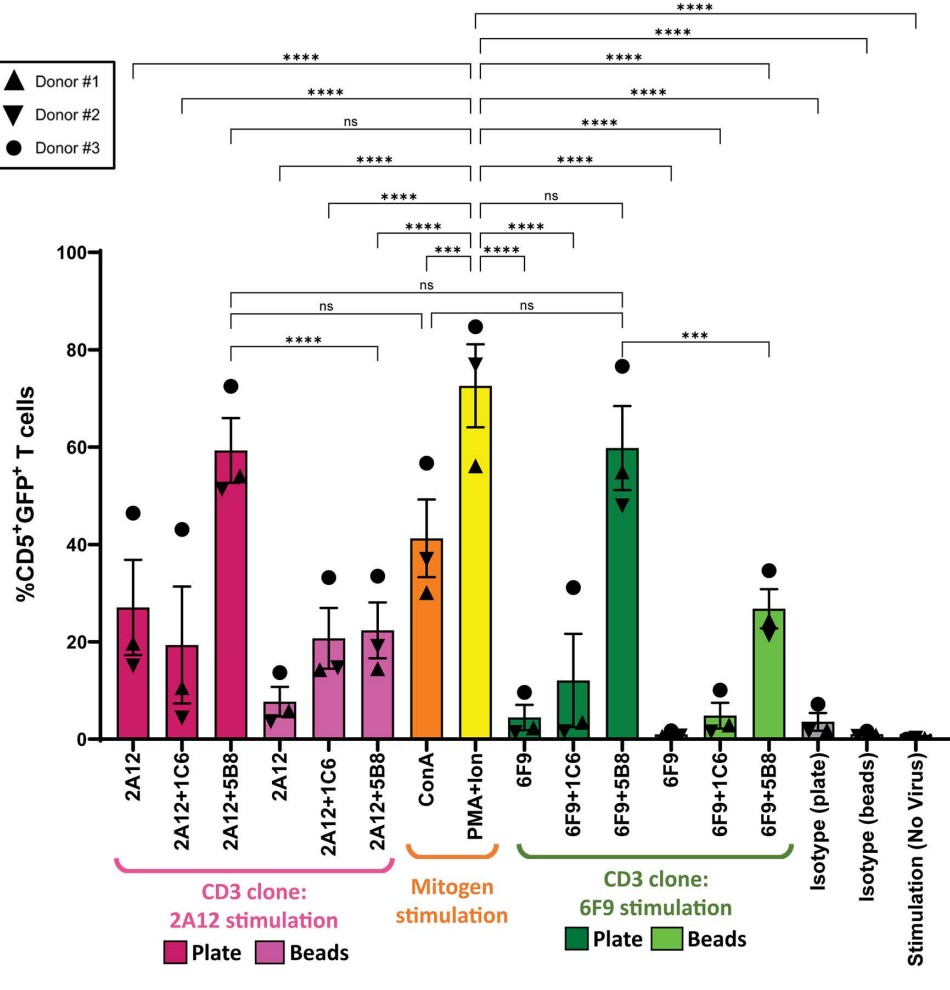

**Fig 6. PMA/I stimulation results in superior T cell transduction efficiency.** PBMCs from three canine donors were transduced after three days of stimulation using mitogen, plate-, or bead-bound antibody (6F9=CA17.6F9; 2A12=CA17.2A12). PBMCs were transduced with GFP gamma retrovirus at an MOI of 13 and the frequency of GFP expressing CD5+ T cells was evaluated three days after transduction. Horizontal lines indicate mean values, and error bars represent standard error of the mean. Pairwise statistical analysis was performed by One-way ANOVA with Tukey's multiple means comparison; *** $p < 0.001$, **** $p < 0.0001$, ns no significance.

### PMA/I stimulation may drive greater incidence of successful T cell-virus interaction

Prior to reintroducing transduced autologous T cells to patients for cancer treatment, T cells are generally expanded *ex vivo* for 10–14-days, over which time frequency of transduced T cells should be maintained. To further explore this, we assessed impact of our four best T cell stimulation strategies on the degree and duration of transgene expression over a 14-day period (Fig 7A). PBMCs from five canine donors (three previous donors with two additional) were stimulated with mitogens or plate-bound 2A12 or 6F9 with 5B8 co-stimulation and transduced once with GFP retrovirus. PBMCs were removed from stimulatory mitogens and antibody, expanded in IL-2 enriched media, and GFP expression was evaluated 4 and 14 days after transduction.

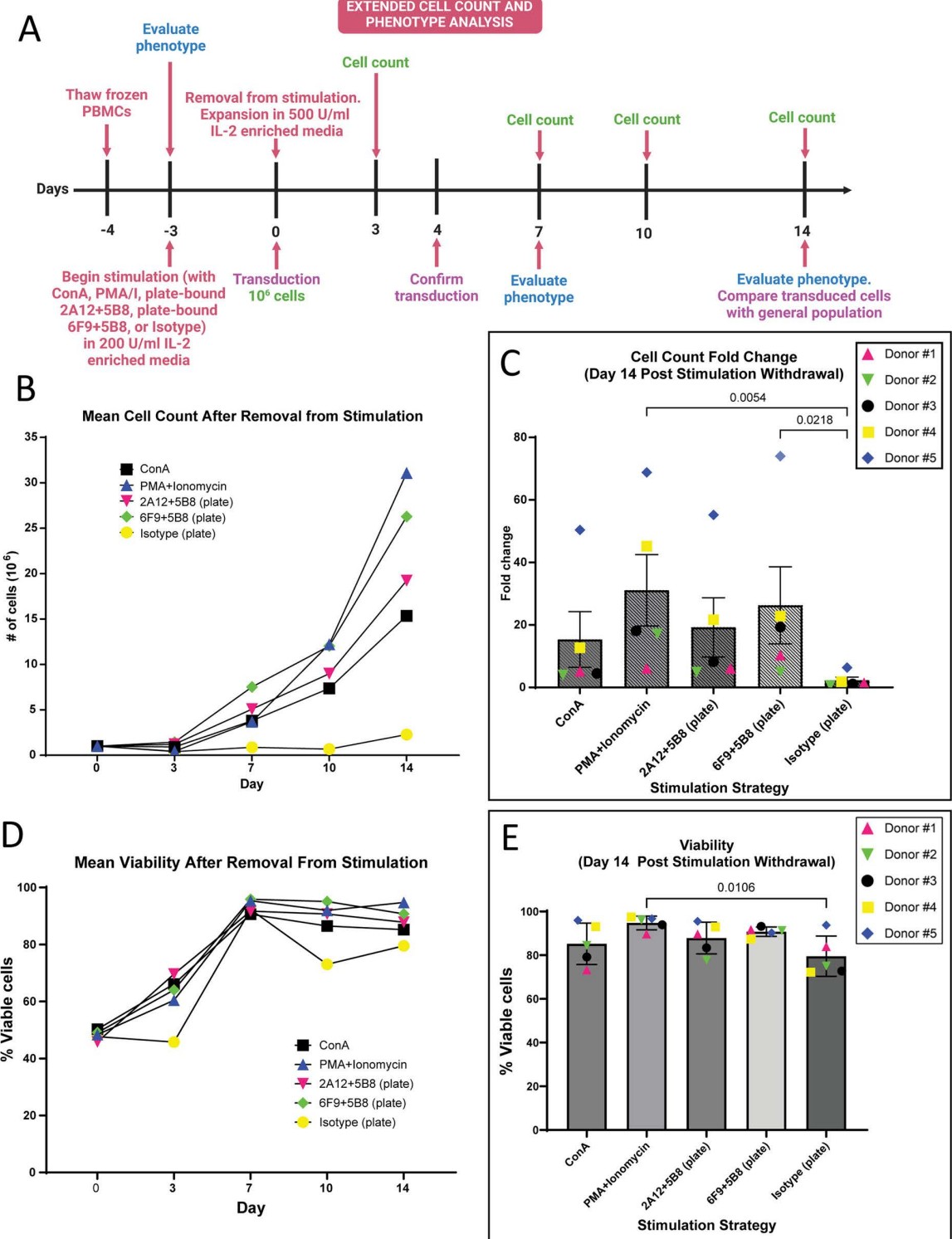

**Fig 7. PMA/I and plate-bound 6F9+5B8 stimulated T cells demonstrate better expansion over 14 days after removal from stimulation. (A)** Schematic representation of experimental design and timeline. After three days of stimulation with mitogen or plate-bound antibody (6F9=CA17.6F9; 2A12=CA17.2A12), PBMCs (n = 5) were removed from stimulation and $10^6$ cells from each sample were expanded in culture. **(B)** Cell count was

performed and the mean count was recorded over 14 days and **(C)** fold change was evaluated on day 14. **(D)** Mean viability was recorded over 14 days with **(E)** statistical analysis of viability performed on day 14. For **(C, E)**, Horizontal lines indicate mean values, and error bars represent standard error of the mean. Pairwise statistical analysis was performed by One-way ANOVA with Tukey's multiple means comparison; $p < 0.05$ is considered significant.

At day 4, the relative trends for all groups were very similar to the previous experiment, though slightly higher transduction rates were achieved. With increased donor numbers, new statistically significant improvement for PMA/I was noted compared to antibody-coated plates (S10A Fig). To better understand the degree of transduction, we also evaluated the median fluorescence intensity of GFP signaling in the transduced T cell population. T cells transduced after PMA/I stimulation emitted a dramatically and significantly higher GFP signal intensity than all other stimulation strategies. GFP signal intensity with PMA/I stimulation was almost 2-fold higher than the next most intense signal from 6F9+5B8 stimulation and almost 2.5-fold higher than ConA and 2A12+5B8 stimulated T cells (S10B Fig). This suggests that PMA/I induced T cell activation may promote a higher degree of successful virus particle and T cell interaction during the transduction process. Evaluation of transduction at 14 days demonstrated that transduced T cells maintained GFP expression after 14 days with no significant decrease in the frequency of transduced T cells (S10C Fig).

### PMA/I and plate-bound 6F9+5B8 induce robust T cell expansion over 14 days

Next, we determined the impact of our four optimal stimulation strategies on expansion over 14 days, which would reflect the *ex vivo* expansion period in a clinical setting. PBMCs from each of the five donors were removed from stimulation after three days and were then transferred to IL-2 enriched media free of stimulatory mitogen or antibody and allowed to expand for 14 days, over which time cell count and viability were evaluated (Fig 7A). After 7 days, plate-bound 6F9+5B8 resulted in superior PBMC expansion over the other stimulation strategies with a greater than 7-fold increase in cell count. PMA/I stimulation resulted in the worst expansion profile at 7 days with an expansion rate of less than half that of 6F9+5B8 stimulated cells (Fig 7B). Surprisingly, PMA/I-induced expansion dramatically improved in the following 7 days, and this stimulation strategy resulted in superior PBMC proliferation over all other strategies by day 14. At day 14, PMA/I stimulated cells had greater than 30-fold increase in mean cell count, compared to approximately 26-fold with plate-bound 6F9+5B8 stimulation. Both strategies showed significant expansion over the unstimulated control (Fig 7B and 7C). Viability after 14 days was above 85% with all stimulation strategies. Highest viability was seen with PMA/I and 6F9+5B8 at 94.8% and 90.7%, respectively, with only PMA/I reaching statistical significance over the unstimulated control (Fig 7D and 7E).

### Plate-bound 6F9+5B8 and ConA reduce the proportion of double negative T cell subsets

The importance of the CD4:CD8 ratio in the T cell infusions for patients receiving immunotherapy for cancer treatment has been highlighted by several previous reports, with a ratio of 1:1 commonly proposed to drive optimal clinical responses [52,53]. To understand how stimulation strategy can impact the prevalence of CD4/CD8 T cell subsets, PBMCs from five donors were stimulated with our four best activation strategies, then expanded in IL-2 enriched media free of stimulatory mitogen or antibodies for 14 days. T cell phenotype was evaluated in CD3$^+$ T cells at day 7 and 14 (Figs 7A, S11A and S11B).

Prior to stimulation, the CD4:CD8 ratio was roughly 4:1 in naive T cells. Without stimulation, the CD4:CD8 ratio at 1:1.5 began to spontaneously approach optimal by day 14. This shift, however, was accompanied by a dramatic increase in the frequency of the CD4$^-$CD8$^-$ double negative population which made up approximately 60% of CD3$^+$ cells (Fig 8). This increase may be attributable to the significantly higher proportion of CD25$^+$ double negative T cells compared CD4 or CD8 single positive T cells in naive populations as reported by Rabiger [22], which would make them more apt to expand in IL-2 enriched media. The trend toward double negative T cell dominance was offset by all stimulation strategies, most

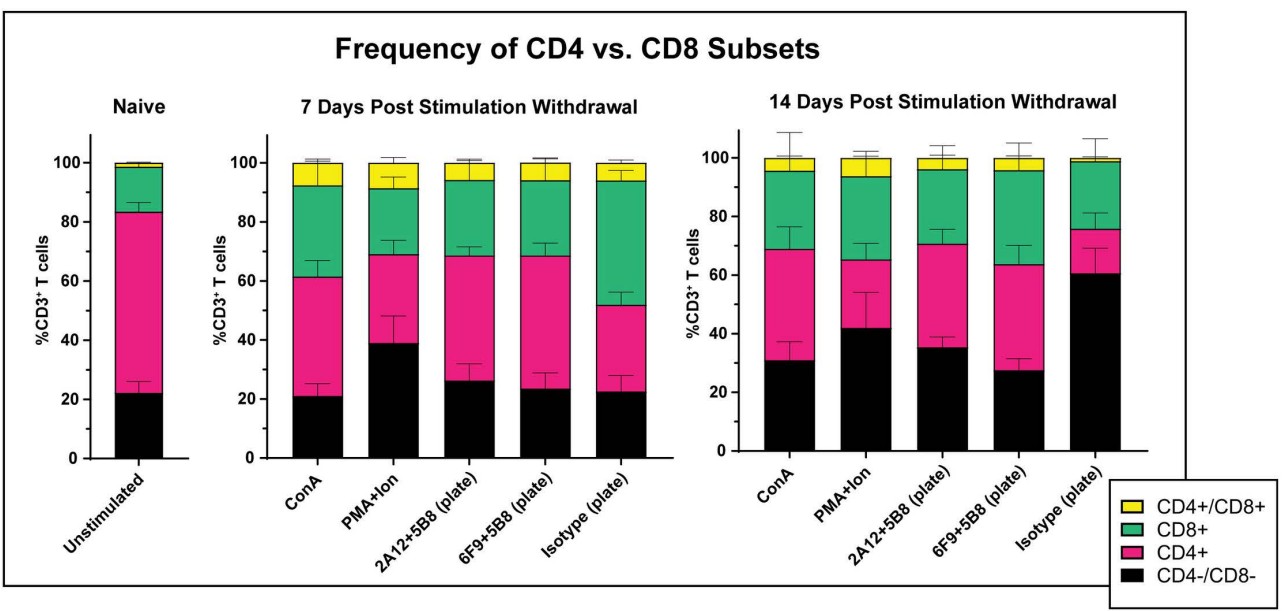

**Fig 8. Plate-bound 6F9 + 5B8 and ConA stimulation reduce proportion of double negative T cells.** After three days of stimulation with mitogen or plate-bound antibody (6F9=CA17.6F9; 2A12=CA17.2A12), PBMCs (n = 5) were removed from stimulation. Cell surface expression of CD4 and CD8 was evaluated before stimulation (naive), and at 7 and 14 days after removal from stimulation. Horizontal lines indicate mean values, and error bars represent standard error of the mean.

notably by ConA and plate-bound 6F9 + 5B8 antibody, which drove a significantly lower frequency of double negative cells while maintaining optimal CD4:CD8 ratios (Figs 8 and S12A).

CD4+CD8+ double positive T cells occurred with low frequency in CD3+ T cells at 14 days with only about 1% present in the unstimulated population and below 5% present with all stimulation strategies. The one exception was PMA/I, which produced slightly higher percentages of double positive cells. Though the proportion was low, PMA/I stimulation did produce significantly more double positive T cells compared to the unstimulated control (S12B Fig).

We next looked at the CD4:CD8 ratios of the transduced cells compared to the general T cell population to determine if phenotype was a factor in transduction propensity. In general, double negative T cells were slightly less prevalent in transduced populations at day 14, which was more apparent with ConA stimulation. Interestingly, this reduction in double negative cells did not notably impact the CD4:CD8 ratios (S13 Fig).

Together, these findings suggest that ConA and plate-bound 6F9 + 5B8 stimulation promote optimal CD4:CD8 ratios while reducing the frequency of double negative T cells. These strategies do not impact double positive T cell frequencies, which are very low.

### PMA/I and 6F9 + 5B8 stimulation reduces the proportion of regulatory T cells

Regulatory T cells ($T_{regs}$) are important in modulating immune response to prevent autoimmune dysfunction. However, their presence in the tumor microenvironment has been shown to negatively impact the effector function of tumor infiltrating lymphocytes (TILs) [54–56]. Additionally, $T_{regs}$ present during adoptive cell transfer for cancer treatment has the potential to negatively impact clinical response [57–59]. Because CD25 is constitutively expressed in $T_{regs}$ as well as induced in activated T cells, it is important to differentiate these populations and to determine how these stimulation strategies impact $T_{reg}$ differentiation. To evaluate this, PBMCs from 5 canine donors were stimulated and expanded as previous (Fig 7A) and the presence of $T_{regs}$ (CD4+CD25+Foxp3+) was evaluated at day 7 and 14 of expansion (S14A and S14B Fig).

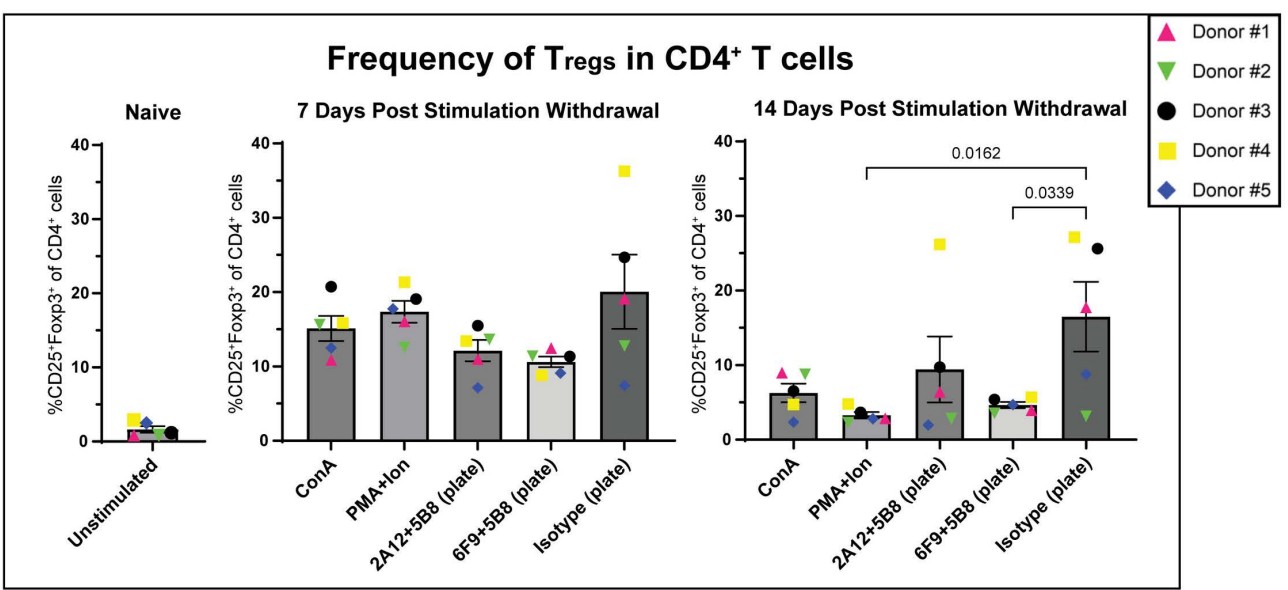

**Fig 9. PMA/I and 6F9+5B8 stimulation significantly reduces the proportion of T_regs.** After three days of stimulation with mitogen or plate-bound antibody (6F9=CA17.6F9; 2A12=CA17.2A12), PBMCs (n=5) were removed from stimulation. Identification of T_regs was determined by surface expression of CD4 and CD25 and intracellular expression of Foxp3. Frequency of T_regs was evaluated before stimulation (naive), and at 7 and 14 days after removal from stimulation. Horizontal lines indicate mean values, and error bars represent standard error of the mean. Pairwise statistical analysis was performed by One-way ANOVA with Tukey's multiple means comparison; $p < 0.05$ is considered significant.

The prevalence of T_regs, made up less than 2% of the naive CD4+ T cell population. Without stimulation, T_reg frequency expanded to 20% after 7 days and reduced slightly to about 16% at day 14 (Fig 9). Naive T cell populations have very low baseline CD25 expression. The constitutive CD25 expression in the T_reg subpopulation was the most likely contributor to disproportionately higher expansion over time in IL-2 enriched media [60], though the relative expansion compared to stimulated groups was quite modest (Fig 7B and 7C).

A similar pattern of T_reg expansion at 7 days and reduction at 14 days was seen with all stimulation strategies, with initial expansion less dramatic with plate-bound antibody stimulation. At 14 days in the case of 6F9+5B8, T_reg differentiation was significantly reduced compared to the unstimulated control. PMA/I stimulation was unique in that it resulted in the highest T_reg expansion at 7 days, but was followed by the most dramatic reduction in frequency at 14 days. At approximately 3%, PMA/I stimulation yielded not only a significantly lower frequency of T_regs than the unstimulated control, but also a lower frequency than all other stimulation strategies. It is important to note that this reduction in T_reg frequency correlated well with the dramatic increase in PBMC expansion seen from days 7–14 in culture (Figs 7B and 9), suggesting that non T_reg populations outpaced T_reg proliferation rates in culture.

To determine how different stimulation strategies affect the prevalence of T_regs in transduced populations, the T_reg proportion in transduced populations were compared to that of the general T cell population at day 14. Frequency of T_reg subsets in transduced T cells closely reflected the phenotypic distribution in the general population with all stimulation strategies, except ConA where T_reg frequency was slightly higher in the transduced population (S15 Fig).

Taken together, these findings suggest that PMA/I and plate-bound 6F9+5B8 stimulation limit T_reg frequency, which may contribute to less negative immunomodulatory effects of the therapeutic T cells.

### Culturing at 38.8°C improves transduction efficiency and expansion

Rotolo et al. noted that increasing culture temperatures from 37°C to 38.8°C resulted in higher rates of transduction and expansion when stimulating with bead-bound 2A12+1C6 [38]. To explore this further in the context of mitogens

and plate-bound antibody, PMBCs from three canine donors were stimulated with PMA/I or plate-bound 2A12+5B8 at 37°C or 38.8°C. The αCD28 clone 1C6 was avoided, as we demonstrated inhibitory effects when 1C6 is plate-bound. After three days, PBMCs were transduced once with GFP-encoding gamma retrovirus and expanded at 37°C or 38.8°C for seven days. GFP expression was evaluated after three days, and cell counts were performed on day 3, 5, and 7 post-transduction.

Increasing culture temperatures to 38.8°C resulted in higher transduction efficiency for all donors. Transduction rates after plate-bound 2A12+5B8 stimulation increased by 6.5% (range 1.5–13.3%) at 38.8°C and by 11.1% (range 6.4–15.4%) with PMA/I stimulation (Fig 10A). Culturing at higher temperatures dramatically improved expansion rates. After 2A12+5B8 stimulation, expansion at 38.8°C resulted in 55-fold expansion compared to 31-fold at 37°C. Expansion at 38.8°C after PMA/I stimulation was even more dramatic, resulting in 115-fold expansion compared to 65-fold at 37°C (Fig 10B and 10C), which reached statistical significance. Furthermore, culturing at 38.8°C abrogated the lag in expansion seen with PMA/I at 37°C (Figs 7B and 10B).

Taken together, these results confirm previous reports that stimulation and transduction at 38.8°C improves transduction (with virus stable at higher temperatures) and also boosts expansion potential for both mitogen and plate-bound antibody stimulation strategies.

## Discussion

Despite multiple approved clinical products for human round cell neoplasia [61], the use of genetically engineered T cells for canine immunotherapy is relatively new. Though positive clinical response in canine safety and clinical trials inspires optimism for CAR T cell therapy as an effective adjunct therapeutic in canine neoplasia [6,14,35], CAR T cell manufacturing in dogs is far from perfected. Optimizing canine T cell activation is an important first step, as robust proliferation that results from activation will ensure adequate patient dosing as well as promote the highest rates of transduction during the CAR T cell manufacturing process [51]. Up to this point, however, investigation into canine T cell activation has been haphazard, with only a few conditions tried in each report and the various approaches never compared to each other. Antibody and mitogen-based stimulation strategies present attractive T cell activation approaches due to their reliability, accessibility, and ease of use. In this study we investigated all four published stimulatory αCD3 and αCD28 antibody clones and directly compared their efficacy in T cell stimulation. αCD3 clones CA17.2A12 (2A12) or CA17.6F9 (6F9) and αCD28 clones 5B8 or 1C6 were either bound to beads or adsorbed to polystyrene plates and were compared to the underexplored T cell stimulatory mitogens PMA/I and ConA in their ability to activate, expand, and transduce canine T cell populations.

In seeking the ideal canine T cell stimulation strategy for patients receiving T cell immunotherapy, we determined that *ex vivo* stimulation with plate-bound 6F9+5B8 or PMA/I resulted in superior activation, long term expansion, viability, and transduction compared to all other strategies evaluated. Additionally, stimulation using these strategies resulted in lower frequency of $T_{regs}$ in the would-be infused T cell population. Taken together, these benefits may contribute to improved treatment efficacy [57–59]. Both strategies drove CD4:CD8 ratios very close to 1:1, though 6F9+5B8 favored a higher CD4 proportion and PMA/I stimulation a higher CD8 proportion, likely attributable to the CD4 downregulatory effects of PMA described previously [62]. A ratio of 1:1 has been reported to be ideal in light of human and mouse immunotherapy publications [52,53], but we should note that the uniquely high frequency of double negative T cells seen in dogs complicates any predictions for clinical outcome derived from these studies.

PMA/I induced higher proportions of CD4⁻CD8⁻ double negative T cell populations than plate-bound 6F9+5B8. Rabiger et al. [22] published a thorough characterization of double negative T cells with a median of 15% of circulating αβ T cells and 40% of circulating γδ T cells expressing neither CD4 or CD8. Of note, dogs are a "γδ T cell low" species, with this subset accounting for approximately 2% of circulating lymphocytes. Experimentally, double negative canine T cells seem to function to some degree similarly to T helper (Th)2 and Th17 cells in their production of cytokines consistent with Type

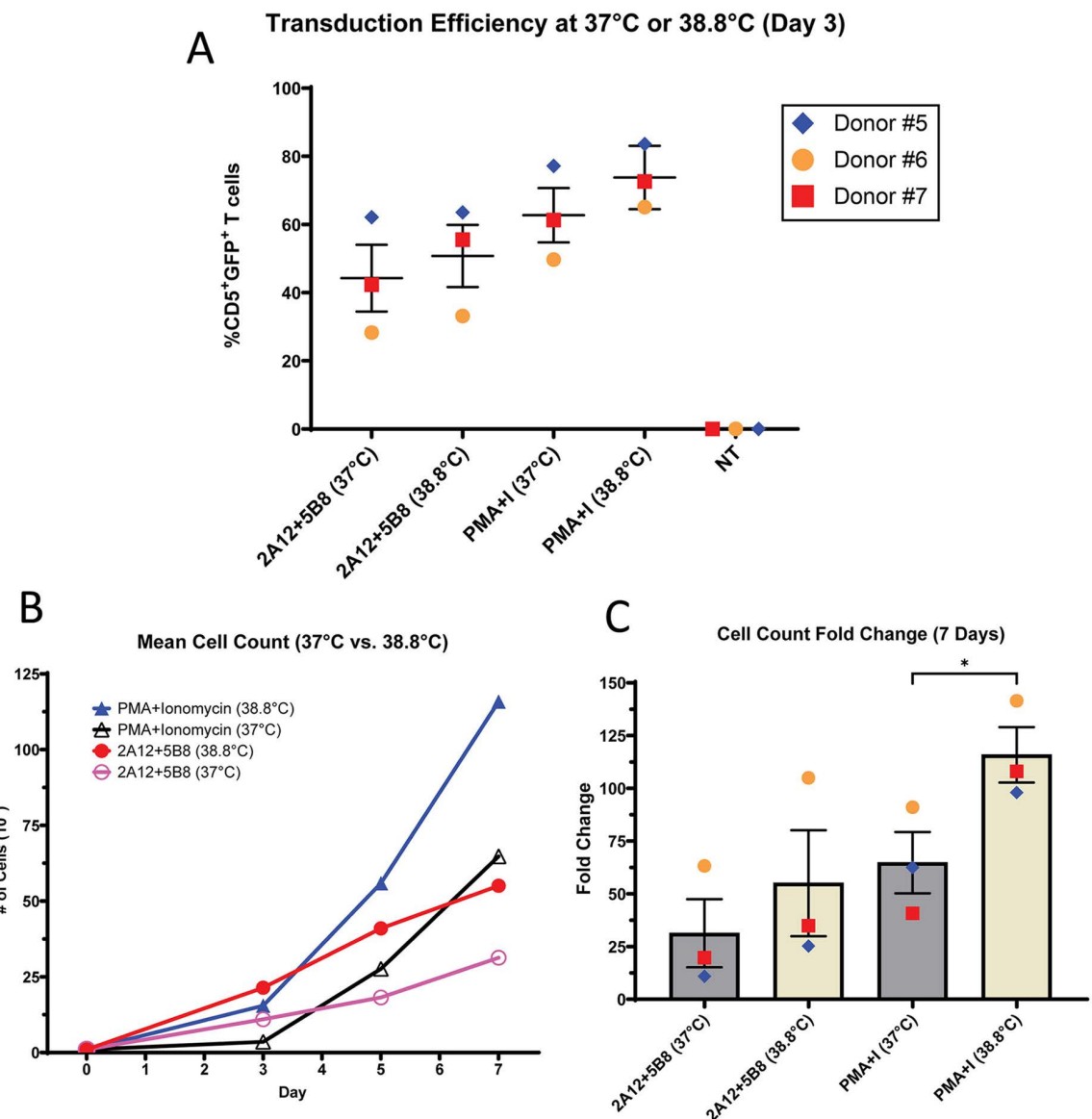

**Fig 10. Culturing at 38.8 °C improves canine T cell transduction and expansion.** After three days of stimulation with PMA/I or plate-bound anti-body (2A12=CA17.2A12), PBMCs (n = 3) were removed from stimulation and $10^6$ cells were transduced with GFP gamma retrovirus. Cells were then expanded in IL-2 enriched media. **(A)** Transduction efficiency was determined by the frequency of CD5+ T cells expressing GFP three days after trans-duction. **(B)** Cell count was performed and the mean count was recorded over seven days and **(C)** fold change was evaluated on day 7. Horizontal lines indicate mean values, and error bars represent standard error of the mean. Pairwise statistical analysis was performed using paired two-tailed t-test; * p < 0.05.

2 and Type 3 pro-inflammatory reactions [24,63], the latter of which may be of clinical benefit [64]. Interestingly, these cells may also have immunosuppressive function, as demonstrated by constitutive expression of IL-10 and mRNA upregulation of *CTLA4* in many of these cells. More investigation is needed to determine the clinical significance of these complex cell types, though continued CD3 stimulation in IL-2 enriched media over the expansion period may markedly reduce the frequency double negative T cells in the event they are deemed detrimental [48]. PMA/I also induced a higher degree of

CD4+CD8+ double positive T cells. However, while Bismarck and others have suggested that double positive T cells may actually repress T cell activation, suggesting an immunoregulatory function [65–67], their low frequency (~ 6%) makes clinical relevance unlikely.

Looking ahead to clinical use, determining whether PMA/I or plate-bound 6F9 + 5B8 should be used will vary based on specific veterinary, academic, and research applications. Mitogen-induced stimulation is attractive for T cell activation because mitogens are readily available, relatively inexpensive, and can be easily stored long term under proper conditions. In the case of PMA/I, our studies demonstrated that this stimulation consistently resulted in the highest transduction compared to all other strategies evaluated, with rates consistently above 70% and the highest rate of 84.7% achieved with a single transduction round. These findings seemed to contradict an earlier report of mitogen-induced transduction, where PMA/I induced transduction did not exceed 26% [17]. Our findings also suggest that PMA/I may promote a higher degree of cell-virus interaction as determined by increased signal MFI in transduced cells. However, confirmation using analysis of vector copy number (VCN) will be necessary in future experiments to verify the extent of transduction events. Confirmation of greater transduction events would suggest that less virus is necessary for the transduction process using PMA/I stimulation, translating to substantial cost reduction.

Unfortunately, mitogens induce T cell activation in a nonspecific manner, so off-target effects may be seen. B cell, phagocyte, and platelet activation are possible with PMA/I [68,69] which contribute to marked inflammatory responses with its use *in vivo* [70]. Though impact on tumor cells is complex [71,72], PMA use *in vivo* has the potential to exacerbate existing neoplastic disease processes [73,74]. These observations drive concerns for using PMA/I-induced stimulation in novel therapeutic strategies that seek *in vivo* T cell activation. For example, recent reports on implantable scaffolds that can induce activation, transduction, expansion, and release of genetically modified T cells [75–78] require *in situ* T cell activation strategies which might not be compatible with mitogen activation. Alternatively, murine studies have successfully demonstrated therapeutic benefit of infused T cells that have been activated *ex vivo* by PMA/I [79], which may translate to successful use in canines as well. While PMA as a known tumor promoter has the potential to induce malignant transformation in T cells [80,81], reports of T cell transformation in human and murine research settings are lacking despite PMA/I's frequent use for *in vitro* T cell stimulation. Though future therapeutic application of PMA/I-induced CAR T cells for neoplasia in dogs is possible, risk of T cell transformation in PMA/I stimulated canine T cells must be thoroughly assessed before clinical trials are attempted. Without further investigation, use of PMA/I-induced CAR T cells will likely be relegated to *in vitro* studies [17,82].

Antibody induced stimulation is advantageous in that the specificity for cell surface CD3 and CD28 induces T cell activation with minimal risk for off-target effects. This also represents the most common activation method for T cell immunotherapy [25–27]. With the evaluation of our two αCD3 antibody clones alone or in combination with our two αCD28 antibody clones on either coated plates or coated beads, we evaluated 12 antibody-mediated stimulation strategies, very few of which have been thoroughly investigated in the literature. In fact, our most potent antibody combination for canine T cell stimulation, 6F9 in combination with 5B8, has never been described in the context of transduction [37]. At lower doses of antibody than that described in other publications [13,21,35], plate-bound 6F9 + 5B8 stimulation resulted in consistent transduction above 50% after a single round, with 77.3% being the highest rate achieved. While similar transduction was seen with plate-bound 2A12 + 5B8, superior expansion and a more desirable T cell phenotype profile was demonstrated by 6F9 + 5B8, leading us to conclude that this combination is best suited for antibody-induced canine T cell stimulation.

In all cases, we saw that plate-bound antibody combinations with 5B8 co-stimulation outperformed their bead-bound counterparts with a greater than two-fold increase in transduction. We considered possible causes of reduced transduction using antibody-conjugated beads. Though beads were removed from samples prior to transduction and no beads were observed grossly even after centrifugation as indicated by their distinct brown color, we considered the possibility that the presence of beads at the microscopic level could interfere with cell-virus attachment at the time of spinoculation. Additionally, it is feasible that incompletely removed beads present in culture could stimulate naive T cells and induce

proliferation during the expansion period after transduction. As naive T cells are unlikely to be transduced, expansion of this population may artifactually dilute GFP+ signaling at the time of analysis. However, we considered this explanation unlikely because bead contamination was not observed during flow cytometry for bead-stimulated samples (S16A–S16E Fig). Even so, this possibility does highlight the importance of quality control measures to ensure bead removal prior to patient infusion [27,83–87]. The need to remove beads, along with the cost to procure the beads, the labor associated with antibody conjugation, and their inferior performance relative to plate-bound antibodies make the continued use of bead-bound antibodies difficult to justify for canine T cell stimulation.

Others groups have described success with different antibody strategies. Zhang et al. [35] used 6F9 for primary stimulation and reported some success utilizing a plate-bound 6F9 + 1C6 strategy. In their hands, this strategy yielded consistently greater than 30% transduction and led to two patients in a safety trial receiving adoptive transfer of CAR T cells with 32.9% and 47.6% transduction efficiency. They observed transduction greater than what we achieved with this antibody strategy. We suspect the discrepancy was mostly likely associated with differences in our stimulation and transduction protocols with their experiments using much higher doses of αCD3 antibody and implementing two rounds of transduction. Other groups have described success with bead-bound 2A12 + 5B8-induced transduction with lentivirus [6] or retrovirus [15]. In the case of the former, co-culture with target cells was necessary to achieve high enough frequency of CAR T cells for canine clinical trials. This strategy was effective, but there is concern that it may prematurely drive the T cell population toward a terminally differentiated effector T cell phenotype, as opposed to a less differentiated memory T cell phenotype, which may interfere with function and persistence long term [88]. In the latter case, retrovirus transduction rates averaged 37%, which was higher than rates we achieved with this strategy. Again, a double transduction protocol was described which may account for the variance. From our studies, we propose that abandoning bead-bound antibodies for a plate-bound method may yield similar or better transduction without the need for two rounds of transduction.

This study has several limitations. First, all transduction experiments utilized GFP gamma retrovirus with GFP serving as the marker for successful transduction. However, the metabolic demands of GFP production differ from those required for the much larger CAR protein, and cell surface CAR expression may not correlate with GFP expression. Second, while transduction efficiency and T cell phenotype serve as predictors for therapeutic potential, they do not necessarily translate to superior functionality. Future work will explore how the optimized stimulation identified here can be harnessed to produce tumor antigen-specific CAR T cells, including functional assays to assess target cell killing, cytokine production, T cell exhaustion, and memory differentiation [89–91].

All experiments in this report were performed using cryopreserved canine PBMCs from healthy canine donors (S1 Table). Primary PBMCs were selected over established canine T cell lines because robust stimulation—a critical element of this research—would not be physiologically relevant in immortalized cell lines. Additionally, since CAR T cell therapies almost always use autologous primary cells, we reasoned that using primary cells from genetically diverse donors strengthened the therapeutic relevance and applicability of findings to the broader veterinary population. Future studies including more breed variety and clinically affected dogs will further enhance translational relevance. Future work could also utilize isolated T cells, which would likely reduce T cell transduction and expansion variability and minimize differences attributable to the inherent cellular heterogeneity of PBMCs.

It is likely that use of fresh PBMCs in future experiments would yield more robust expansion and transduction than cryopreserved cells. Expansion may also be limited by the anticoagulant utilized at the time of blood collection. CPDA-1 has been shown to reduce human lymphocyte proliferation in response to stimulation [92]. While it is unclear how CPDA-1 affects canine T cells, utilizing heparin as an anticoagulant for future blood products may improve proliferation rates. Finally, in this report, we tested transduction after three days of activation. Although 3-day transduction is common in the canine literature [13,15,17], others have explored transduction at 24 and 48 hours after activation [6,38]. Having established optimal stimulation strategies, future work can look into optimizing transduction.

Future work will also seek to establish and maintain a substantial stem cell memory T cell ($T_{SCM}$) population after stimulation. Adoptive transfer of less differentiated memory T cell subsets has been linked to better persistence and tumor regression in both human trials and mouse models with genetically modified T cells. This has been attributed to the less differentiated $T_{SCM}$ population's replenishment of the more differentiated central memory ($T_{CM}$), effector memory ($T_{EM}$) and effector ($T_{EFF}$) T cell subsets for the duration of the anti-tumor response [89–91]. Cieri [93] demonstrated that the presence of IL-7 and IL-15 in culture promotes survival and expansion of human T cells while promoting a less differentiated $T_{SCM}$ memory phenotype. This has also been demonstrated in canine T cells to some degree by Panjwani [6], who demonstrated cross reactivity of human recombinant IL-7 and IL-15 in producing less differentiated T cells, though identification was limited to CD62L or CD27 expression, which can identify less differentiated memory phenotypes but cannot alone specifically identify $T_{SCM}$ cells. Sakai explored canine T cell expansion with various combinations of IL-2, IL-15, and IL-21. Though the addition of these cytokines drove more reliable expansion in PHA stimulated T cells with less variability, they failed to produce a substantial proportion of $T_{SCM}$ cells and resulted in predominantly $T_{EFF}$ cell subset differentiation [13]. Further experiments are necessary to fully understand how IL-2, IL-7, and IL-15 influence the repertoire of canine T cell subsets. More recently, nanoparticles utilized for human T cell activation have been shown to influence memory T cell formation [94]. These have been evaluated in the context of a hydrogel delivery mechanism to produce less differentiated CAR T cells for solid tumor treatment [95]. Implantable alginate scaffolds containing IL-2 and stimulatory antibodies [75] have also contributed to CAR T cell persistence *in vivo* as have injectable CAR enhancer proteins that selectively induce IL-2 signaling in CAR T cells [96]. These can be explored in canine T cells as well.

While additional work remains to achieve GMP compliance for the 6F9 and 5B8 antibodies, these findings lay critical groundwork for preclinical and clinical trials in dogs with cancers. Such studies will enable comprehensive assessment of CAR T cell safety, expansion, and therapeutic function *in vivo*. The protocols described here could enhance both the feasibility and accessibility of CAR T cell therapy in veterinary oncology. Importantly, canine studies are increasingly valuable translational bridges to human immunotherapy. Compared to murine studies, dogs with spontaneous disease provide a more immunologically relevant and heterogeneous model. This comparative oncology approach will not only strengthen the translational research pipeline but also will accelerate development of next-generation human therapies.

We hope that the results from this study can be used to promote optimal conditions for canine T cell transduction. We hope that is can bolster CAR T cell frequencies after the implementation of current strategies, such as multiple transduction events or co-culture with target cells, or perhaps eliminate the need for these strategies altogether. We hope this investigation will serve as a framework for those interested in CAR T cell development for use in dogs, both as veterinary patients and as models for human disease.

## Supporting information

**S1 Table. Donor information.** A summary of pertinent medical information for the seven canine blood donors used in this study.
(TIF)

**S1 Fig. Determination of cell viability.** Representative flow cytometry dot plot highlighting gating strategy used to determine cell viability. The parent gate includes all events with the exclusion of debris, followed by gating on singlets, followed by gating on viable cells as indicated by the exclusion of viability dye.
(TIF)

**S2 Fig. Flow cytometry was utilized to verify successful antibody-bead conjugation.** MACSiBeads contain anti-biotin antibody that will bind to biotinylated stimulatory antibody during the conjugation process. Beads were stained using streptavidin-PE to label biotinylated antibody on the surface of successfully conjugated beads. The unstained sample contains

no streptavidin and was used as a negative control. Unconjugated beads contain no stimulatory antibody and thus show no PE signaling. All other beads contain stimulatory antibody and show PE signaling demonstrating successful conjugation.
(TIF)

**S3 Fig. Plate-bound co-stimulatory 1C6 antibody negatively impacts T cell activation.** (A-B) This experiment was performed to confirm the findings in Fig 3 that demonstrated that plate-bound αCD28 clone 1C6 reduced activation efficiency in T cells when used (A) in combination with αCD3 clone 2A12 compared to stimulation with (B) 2A12 alone (2A12=CA17.2A12). Plate-bound stimulatory antibodies were used to activate PBMCs (n = 3) and frequency of CD5$^+$ T cell activation was evaluated by cell surface expression of CD25. Horizontal lines indicate mean values, and error bars represent standard error of the mean. Pairwise statistical analysis was performed by One-way ANOVA with Tukey's multiple means comparison; * $p < 0.05$, ** $p < 0.01$, *** $p < 0.001$, **** $p < 0.0001$.
(TIF)

**S4 Fig. Co-stimulation alone does not induce T cell activation.** (A-B) Frequency of CD5$^+$ T cell activation in three canine donors as indicated by cell surface expression of CD25. Plate-bound αCD28 antibody (A) 1C6 or (B) 5B8 of different doses was used to determine potential for T cell stimulation when used in the absence of αCD3 antibody. Pairwise statistical analysis was performed by One-way ANOVA with Tukey's multiple means comparison; **** $p < 0.0001$.
(TIF)

**S5 Fig. Antibody dosage increase does not promote AICD.** Representation of the effect of dose increase of stimulatory antibody on cell viability. Plate-bound (A) 6F9+5B8 and (B) 2A12+5B8 were used to stimulate PBMCs from three donors (6F9=CA17.6F9; 2A12=CA17.2A12). On day 3 viability was evaluated by flow cytometry as the frequency of viable singlets. Less than 10% loss in viability was considered acceptable and is indicated by green arrows. Horizontal lines indicate mean values, and error bars represent standard error of the mean.
(TIF)

**S6 Fig. Evaluation of canine T cell activation.** (A) Gating strategy for identifying activated T cells. Lymphocytes are gated in SSC/FSC plot followed by exclusion of doublets. Next, gating on viable cells by exclusion of dead with Live-or-Dye 405/452 viability dye. T cells were identified by gating on CD5$^+$ cells followed by gating on CD25$^+$ cells to identify activated T cells. (B) Representative dot plots for all stimulation strategies are shown (three days post-activation).
(TIF)

**S7 Fig. Plate-bound αCD3 with 5B8 co-stimulation results in better activation than bead-bound antibody.** Summary of optimal doses of stimulatory antibodies (6F9=CA17.6F9; 2A12=CA17.2A12) used to activate PBMCs (n = 3); frequency of activated CD5$^+$ T cells was determined by cell surface expression of CD25 after three days. This figure contains the same data from Fig 4 with focused statistical analysis on the effectiveness of antibody clones bound to plates or to beads. Horizontal lines indicate mean values, and error bars represent standard error of the mean. Multiple comparison statistical analysis was performed by Two-way ANOVA with Tukey's multiple means comparison; *** $p < 0.001$, **** $p < 0.0001$, ns not significant.
(TIF)

**S8 Fig. IL-2 alone is insufficient to activate canine T cells.** PBMCs from three canine donors were cultured either without stimulation in different concentrations of IL-2 or with stimulation using plate-bound stimulatory antibodies (6F9=CA17.6F9) with or without exogenous IL-2. Unstimulated PBMCs cultured in media with no IL-2 were used as negative controls. Frequency of activated CD5$^+$ T cells was determined by cell surface expression of CD25 at day 3 and 5. Horizontal lines indicate mean values, and error bars represent standard error of the mean.
(TIF)

**S9 Fig. Plate-bound αCD3 with 5B8 co-stimulation results in better transduction efficiency than bead-bound antibody.** After three days of antibody-induced stimulation (6F9=CA17.6F9; 2A12=CA17.2A12), PBMCs from canine donors (n = 3) were transduced with GFP gamma retrovirus. Frequency of GFP expressing CD5+ T cells was evaluated three days after transduction. This figure contains the same data from Fig 6 with focused statistical analysis on the effectiveness of antibody clones bound to plates or to beads. Horizontal lines indicate mean values, and error bars represent standard error of the mean. Multiple comparison statistical analysis was performed by Two-way ANOVA with Tukey's multiple means comparison; *** $p < 0.001$, **** $p < 0.0001$, ns not significant.
(TIF)

**S10 Fig. Evaluation of GFP expression, quantification, and duration after mitogen and plate-bound antibody stimulation and transduction.** After stimulation with mitogen or plate-bound antibody (6F9=CA17.6F9; 2A12=CA17.2A12), PBMCs (n = 5) were transduced with GFP gamma retrovirus at MOI of 10. (A) Transduction efficiency was determined by the frequency of CD5+ T cells expressing GFP four days after transduction. (B) The intensity of GFP signaling using median fluorescence intensity (MFI) in CD5+GFP+ T cells was used as an indicator of virus particle/cell interaction. (C) To determine persistence of transduced T cells, frequency of GFP+CD5+ T cells was evaluated at day 14 and compared to day 4. Horizontal lines indicate mean values, and error bars represent standard error of the mean. Pairwise statistical analysis was performed by One-way ANOVA with Tukey's multiple means comparison; ** $p < 0.01$, *** $p < 0.001$, **** $p < 0.0001$, ns no significance.
(TIF)

**S11 Fig. Evaluation of canine CD4/CD8 T cell subsets.** (A) Gating strategy for identifying CD4 and CD8 T cell subsets. Lymphocytes are gated in SSC/FSC plot followed by exclusion of doublets. Next, gating on viable cells by exclusion of dead with LIVE/DEAD Near IR viability dye. T cells were identified by gating on CD3+ cells. Finally, CD4 and CD8 expression were evaluated. Phenotype was determined according to quadrant: Upper left quadrant – CD4+ cells; upper right – double positive cells; lower left – double negative cells; and lower right – CD8+ cells. (B) Representative dot plots for each stimulation strategy are shown (14 days after removal from stimulation).
(TIF)

**S12 Fig. Plate-bound 6F9+5B8 and ConA significantly reduce double negative T cell frequency while PMA/I significantly increases double positive T cells.** After three days of stimulation with mitogen or plate-bound antibody (6F9=CA17.6F9; 2A12=CA17.2A12), PBMCs (n = 5) were removed from stimulation. Cell surface expression of CD4 and CD8 was evaluated before stimulation (naive), and at 7 and 14 days after removal from stimulation. On day 14, frequency of (A) CD4-CD8- double negative T cells and (B) CD4+CD8+ double positive T cells were characterized. Horizontal lines indicate mean values, and error bars represent standard error of the mean. Pairwise statistical analysis was performed by One-way ANOVA with Tukey's multiple means comparison; $p < 0.05$ is considered significant.
(TIF)

**S13 Fig. CD4:CD8 ratio is similar in transduced T cells and the general T cell population.** After three days of stimulation with mitogen or plate-bound antibody (6F9=CA17.6F9; 2A12=CA17.2A12), PBMCs (n = 5) were removed from stimulation and transduced with GFP gamma retrovirus. CD4 and CD8 expression was compared on day 14 between the general CD3+ T cell population and transduced GFP+CD3+ T cells. CD4:CD8 ratio was recorded for the general population (white) and the transduced population (green). Horizontal lines indicate mean values, and error bars represent standard error of the mean.
(TIF)

**S14 Fig. Evaluation of canine T_regs.** (A) Gating strategy for identifying regulatory T cells. Lymphocytes are gated in SSC/FSC plot followed by exclusion of doublets. Next, gating on viable cells by exclusion of dead with LIVE/DEAD Near IR viability dye. CD4+ T cells were next identified. Finally, CD25 surface expression and intracellular Foxp3 were evaluated.

Regulatory T cells were identified as CD25⁺Foxp3⁺ cells in the upper right quadrant. (B) Representative dot plots for each stimulation strategy are shown (14 days after removal from stimulation).
(TIF)

**S15 Fig. T$_{reg}$ frequency is similar in transduced T cells and the general T cell population.** After three days of stimulation with mitogen or plate-bound antibody (6F9=CA17.6F9; 2A12=CA17.2A12), PBMCs (n=5) were removed from stimulation and transduced with GFP gamma retrovirus. On day 14, the frequency of T$_{regs}$ was compared between the general CD4⁺ T cell population and transduced GFP⁺CD4⁺ T cells. Horizontal lines indicate mean values, and error bars represent standard error of the mean.
(TIF)

**S16 Fig. Bead contamination is not apparent during flow cytometry.** (A) Flow cytometry color dot plot demonstrating bead contamination (green dots within the red circle) in an unstimulated lymphocyte population (pink dots) where the beads were intentionally not removed. (B-C) compares a representative (B) plate-bound 6F9+5B8 antibody stimulated (bead free) PBMC population with a representative (C) bead-bound 6F9+5B8 antibody stimulated PBMC population from this study. (D-E) compares a representative (D) plate-bound 2A12+5B8 antibody stimulated (bead free) PBMC population with a representative (E) bead-bound 2A12+5B8 antibody stimulated PBMC population from this study.
(TIF)

## Acknowledgments

We are grateful to Dr. Peter Moore (UC Davis) for gifting the αCD3 antibody (clone CA17.6F9) and to Kristy Harmon for all her assistance; Brian Hayes (Fred Hutchinson Cancer Center) for gifting the 5B8 hybridoma cell line; Dr. Gianpietro Dotti (UNC) for gifting the GFP gamma retrovirus; and Dr. Christopher DeRenzo, Dr. Stephen Gottschalk, and Carla O'Reilly (St. Jude Children's Research Hospital) for their time and support in discussing canine T cell transduction. Additionally, we thank NCSU Laboratory Animal Resources veterinary medical team and North American Veterinary Blood Bank for their collection of canine whole blood and David Rose of the Flow Cytometry and Cell Sorting Core (NCSU) for his support in flow cytometry data collection.

## Author contributions

**Conceptualization:** Treyvon W. Davis, Paul R. Hess, Christopher L. Mariani, Yevgeny Brudno.

**Data curation:** Treyvon W. Davis.

**Formal analysis:** Treyvon W. Davis, Yevgeny Brudno.

**Funding acquisition:** Paul R. Hess, Christopher L. Mariani, Yevgeny Brudno.

**Investigation:** Treyvon W. Davis, Arissa He.

**Methodology:** Treyvon W. Davis, Jennifer C. Holmes.

**Resources:** Jennifer C. Holmes, Paul R. Hess.

**Supervision:** Christopher L. Mariani, Yevgeny Brudno.

**Writing – original draft:** Treyvon W. Davis, Yevgeny Brudno.

**Writing – review & editing:** Treyvon W. Davis, Jennifer C. Holmes, Paul R. Hess, Christopher L. Mariani, Yevgeny Brudno.

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
