## [Decision Letter · Decision Letter 0]

15 Jul 2025

PONE-D-25-20434Optimizing canine T cell activation, expansion, and transductionPLOS ONE

Dear Dr. Brudno,

Thank you for submitting your manuscript to PLOS ONE. After careful consideration, we feel that it has merit but does not fully meet PLOS ONE’s publication criteria as it currently stands. Therefore, we invite you to submit a revised version of the manuscript that addresses the points raised during the review process.

Please respond to reviewers' comments individually.

We look forward to receiving your revised manuscript.

Kind regards,

Xiaosheng Tan

Academic Editor

PLOS ONE

Journal Requirements:

[This work was supported by the National Institutes of Health through Grant Award R37-CA260223 and R33-CA281875 from the NCI, seed funding from the Comparative Medicine Institute and the Lineberger Cancer Center and by start-up funds provided by the University of North Carolina at Chapel Hill, North Carolina State University at Raleigh, and the Lineberger Cancer Center.].

[This work was supported by the National Institutes of Health through Grant Award R37-CA260223 and R33-CA281875 from the NCI, seed funding from the Comparative Medicine Institute and the Lineberger Cancer Center and by start-up funds provided by the University of North Carolina at Chapel Hill, North Carolina State University at Raleigh, and the Lineberger Cancer Center. We are grateful to Peter Moore (UC Davis) for gifting the αCD3 antibody (clone CA17.6F9); Brian Hayes (Fred Hutchinson Cancer Center) for gifting the 5B8 hybridoma cell line; Gianpietro Dotti (UNC) for gifting the GFP gamma retrovirus and Drs. Christopher DeRenzo, Dr. Stephen Gottschalk, and Carla O’Reilly (St. Jude Children’s Research Hospital for their time and support in discussing canine T cell transduction. Additionally, we thank NCSU Laboratory Animal Resources veterinary medical team and North American Veterinary Blood Bank for their collection and donation of canine whole blood and David Rose (NCSU) for his support in flow cytometry data collection.]

[This work was supported by the National Institutes of Health through Grant Award R37-CA260223 and R33-CA281875 from the NCI, seed funding from the Comparative Medicine Institute and the Lineberger Cancer Center and by start-up funds provided by the University of North Carolina at Chapel Hill, North Carolina State University at Raleigh, and the Lineberger Cancer Center.]

[The authors declare the following financial interests/personal relationships which may be considered as potential competing interests: Yevgeny Brudno has patents related to CAR T cell production and therapy pending to NC State. Yevgeny Brudno is the scientific founder of Persistence Therapeutics, which seeks novel methods for CAR T cells therapy. None of the other authors have known competing financial interests or personal relationships that could have appeared to influence the work reported in this paper].

Reviewers' comments:

Reviewer's Responses to Questions

**Comments to the Author**

1. Is the manuscript technically sound, and do the data support the conclusions?

Reviewer #1: Yes

Reviewer #2: Yes

Reviewer #3: Yes

Reviewer #4: Yes

2. Has the statistical analysis been performed appropriately and rigorously? 

Reviewer #1: Yes

Reviewer #2: Yes

Reviewer #3: Yes

Reviewer #4: Yes

3. Have the authors made all data underlying the findings in their manuscript fully available?

Reviewer #1: Yes

Reviewer #2: Yes

Reviewer #3: Yes

Reviewer #4: Yes

4. Is the manuscript presented in an intelligible fashion and written in standard English?

Reviewer #1: Yes

Reviewer #2: Yes

Reviewer #3: Yes

Reviewer #4: Yes

5. Review Comments to the Author

Reviewer #1: This manuscript presents a systematic and detailed study on the optimization of canine T cell activation, expansion, and lentiviral transduction. Given the emerging role of dogs as translational models for human cancer immunotherapy, the work addresses a timely and important challenge: the lack of standardized and effective T cell activation protocols in canines. The authors evaluate different αCD3 and αCD28 antibody clones using bead- and plate-bound formats, compare their performance to mitogens, and assess IL-2’s role in canine T cell biology. They identify plate-bound αCD3 clone 6F9 or 2A12 with αCD28 clone 5B8 co-stimulation as the most effective strategy for activation and expansion, providing a valuable framework for canine adoptive T cell therapy development.

The data are generally well-conceived, and the experiments are thorough and replicable. However, the manuscript would benefit significantly from improved organization, more robust statistical treatment, and a deeper discussion of clinical and translational implications.

1.The introduction and abstract lack a clearly stated central hypothesis or specific research aims.

Recommendation: State explicitly in the introduction whether the goal is to identify an optimal T cell activation protocol, compare bead vs. plate systems, or evaluate mitogens in the context of gene therapy. This framing should guide the manuscript structure.

2.The discussion would benefit from elaboration on how these findings might be used in:

Preclinical trials in dogs,

Companion animal cancer therapy,

Bridging to human T cell engineering.

Recommendation: Add a focused paragraph in the discussion on future directions and clinical translation, including whether GMP-compliant reagents (or alternatives) for 5B8/6F9 are available or needed.

3.CD25 is a useful early activation marker but provides limited insight into T cell functionality.

Recommendation: Include additional activation or functional readouts such as CD69, cytokine production (e.g., IFN-γ, IL-2), or cytotoxic markers to better validate the functional status of activated cells.

4.The manuscript concludes that 5B8 is superior to 1C6, but does not explore why. Similarly, the antagonistic effect of plate-bound 1C6 is described but not investigated.

Recommendation: Consider brief experiments or discussion on possible mechanisms (e.g., differences in epitope binding, receptor internalization, or Fc-mediated effects).

Reviewer #2: Treyvon W. Davis et al. systematically evaluated various stimulation strategies to optimize activation, expansion, and retroviral transduction of canine T cells for immunotherapeutic applications. The study demonstrates that plate-bound antibodies significantly outperform bead-bound antibodies, and that stimulation with the αCD3 clone CA17.6F9 combined with αCD28 clone 5B8—or with the mitogen PMA plus ionomycin—yields superior T cell activation, expansion, transduction efficiency, and phenotypic profiles. These findings offer a valuable framework for advancing canine CAR T cell therapies, with implications for both veterinary medicine and translational human immunotherapy research. I have the following comments and questions:

1. I recommend that the authors provide a more detailed background on the two αCD3 clones (CA17.2A12 and CA17.6F9) and the two αCD28 clones (1C6 and 5B8) in the Introduction or Methods section. This would help readers better understand the rationale behind selecting and comparing these antibody combinations.

2. The study relies on CD25 expression to assess T cell activation. The authors should clarify why CD25 was chosen over the more commonly used early activation marker CD69, and whether any comparative data were available.

3. The abbreviations "2A12" and "6F9" are first introduced at line 379. For clarity and consistency, I suggest defining these abbreviations earlier in the manuscript, ideally when the antibodies are first mentioned.

4. For improved readability, I recommend moving all figure legends to the end of the manuscript, rather than interspersing them within the main text.

5. The study exclusively used fresh or thawed primary canine PBMCs rather than established T cell lines. I encourage the authors to explicitly state this in the Discussion section and to comment on the benefits and limitations of using primary cells—particularly in terms of reproducibility, cell heterogeneity, and functional variability.

6. In Figure S8, I recommend showing individual data points to provide better visual representation of biological variability.

Reviewer #3: In this manuscript, the authors present an extensive and well-executed screening of optimal conditions for canine T cell transduction. Their study includes twelve antibody stimulation strategies, two previously untested mitogens, multiple αCD3/αCD28 combinations, and a comparison of the two most widely used antibody presentation formats. The methodology, results, figure legends, and discussion are all presented with a high degree of clarity, logic, and scientific rigor. Overall, this work makes a valuable contribution to the field.

There are, however, a few formatting and presentation issues that should be addressed to further improve the quality and clarity of the manuscript. My comments are as follows:

1) Line 133. The formatting of “x” in the phrase preceding “DPBS” appears incorrect. Please review and standardize the formatting of this term throughout the manuscript.

2) Line 135. The format used to state centrifugation speed is incorrect. Please check all instances where centrifugation speed is stated and revise for consistency.

3) Line 349. There’s an unusual blank space between lines 349 and 350.

4) Fig. 1. In the legend of Fig. 1, the authors state that * p < 0.05 is used, but such a statistical significance marker is not shown in panels A or C.

5) Fig. 4. Statistics. While one-way ANOVA with Tukey’s multiple comparisons is statistically valid for multiple group comparisons, a Two-way ANOVA accounting for antibody clone and stimulation method would better reflect the factorial design of this experiment.

6) Fig. 6. The same statistical analysis approach used in Fig. 4 should also be applied to Fig. 6. This recommendation also applies to Fig. S8A and S8B.

7) Fig. S13. In the isotype group, the percentage of CD25+Foxp3+ cells appears unusually high at 17.76%. The authors should clarify whether this reflects high background staining or a gating issue. Additionally, in the 2A12+5B8 group, the percentage of CD25+Foxp3+ cells is reported as 9.76%; however, the total number of events in this plot appears markedly lower than in other groups.

Reviewer #4: Yevgeny et al. conducted a valuable investigation into optimizing canine T cell activation, expansion, and transduction. While the study provides a solid foundational framework, additional rigorous and translational studies are necessary to substantiate the proposed mechanisms and enhance the clinical relevance of the findings. Incorporating the following elements would not only reinforce the central hypothesis but also significantly elevate the scientific impact and translational potential of the work.

Major Comments:

1. To deepen mechanistic insight and enhance translational relevance, further analysis of intracellular signaling pathways activated by different stimulation strategies—such as plate-bound 6F9+5B8 antibodies, PMA/ionomycin, or ConA—is highly recommended. This can be achieved through phospho-flow cytometry or Western blotting, focusing on key signaling molecules such as pZAP70, pAKT, and pSTAT5. Complementary transcriptomic profiling using bulk or single-cell RNA sequencing would help define T cell activation states, exhaustion signatures, and transduction susceptibility. Additionally, cytokine profiling via ELISA or Luminex could further elucidate functional polarization and cytokine secretion patterns.

2. To improve post-stimulation proliferation and gene transfer efficiency, a systematic evaluation of supportive cytokines—including IL-2, IL-7, IL-15, and IL-21—should be conducted to optimize T cell expansion while preserving desirable phenotypes. Further process optimizations such as spinoculation, retronectin-coated plates, serum-free culture conditions, and modified temperature regimens (e.g., transduction at 32–34 °C followed by expansion at 38.8 °C) may also enhance viral transduction and cell viability.

3. Long-term functional validation is critical. Incorporating cytotoxicity assays—such as chromium release or LDH assays—using relevant canine tumor cell lines would directly assess the anti-tumor efficacy of the genetically modified T cells. Concurrently, longitudinal flow cytometric analysis of memory markers (CD62L, CCR7) and exhaustion markers (PD-1, LAG-3, TIM-3) would provide insights into the persistence, functional status, and therapeutic durability of the expanded T cell populations.

4. Investigation into canine-specific viral restriction factors such as SAMHD1 and APOBEC3, and the use of small molecule inhibitors or siRNA-mediated knockdown strategies, may further improve retroviral gene transfer efficiency. Moreover, the application of epigenetic modulators, including HDAC or DNA methyltransferase inhibitors, could enhance transgene expression and stability. Together, these mechanistic and translational strategies would not only strengthen the therapeutic foundation of canine T cell immunotherapy but also offer valuable insights for the development of comparable approaches in human cancer immunotherapy.

6. PLOS authors have the option to publish the peer review history of their article (what does this mean?). If published, this will include your full peer review and any attached files.

Reviewer #1: No

Reviewer #2: No

Reviewer #3: No

Reviewer #4: No

---

## [Author Response · Author response to Decision Letter 1]

13 Aug 2025

We sincerely thank all four reviewers for their thorough and constructive evaluation of our manuscript. We are pleased that all reviewers unanimously agreed that our work is technically sound with data that supports our conclusions, that our statistical analysis was performed appropriately and rigorously, that we have made all underlying data fully available in accordance with PLOS policies, and that our manuscript is presented in an intelligible fashion with clear, standard English. This consensus among the reviewers validates our systematic approach to optimizing canine T cell activation, expansion, and transduction strategies, and we appreciate their recognition of the scientific rigor and clarity of our work.

Please note the attached full Reviewer Response document is at the end of PDF. PLOS does not allow it to be moved to the front.

---

## [Decision Letter · Decision Letter 1]

24 Aug 2025

Optimizing canine T cell activation, expansion, and transduction

PONE-D-25-20434R1

Dear Dr. Brudno,

We’re pleased to inform you that your manuscript has been judged scientifically suitable for publication and will be formally accepted for publication once it meets all outstanding technical requirements.

Kind regards,

Xiaosheng Tan

Academic Editor

PLOS ONE

Additional Editor Comments (optional):

Reviewers' comments:

Reviewer's Responses to Questions

**Comments to the Author**

1. If the authors have adequately addressed your comments raised in a previous round of review and you feel that this manuscript is now acceptable for publication, you may indicate that here to bypass the “Comments to the Author” section, enter your conflict of interest statement in the “Confidential to Editor” section, and submit your "Accept" recommendation.

Reviewer #1: All comments have been addressed

Reviewer #2: All comments have been addressed

Reviewer #3: All comments have been addressed

Reviewer #4: All comments have been addressed

2. Is the manuscript technically sound, and do the data support the conclusions?

Reviewer #1: Yes

Reviewer #2: Yes

Reviewer #3: Yes

Reviewer #4: Yes

3. Has the statistical analysis been performed appropriately and rigorously? 

Reviewer #1: Yes

Reviewer #2: Yes

Reviewer #3: Yes

Reviewer #4: Yes

4. Have the authors made all data underlying the findings in their manuscript fully available?

Reviewer #1: Yes

Reviewer #2: Yes

Reviewer #3: Yes

Reviewer #4: Yes

5. Is the manuscript presented in an intelligible fashion and written in standard English?

Reviewer #1: Yes

Reviewer #2: Yes

Reviewer #3: Yes

Reviewer #4: Yes

6. Review Comments to the Author

Reviewer #1: I appreciate the authors’ detailed and thoughtful responses to the initial comments and commend the substantial improvements made in this revised version of the manuscript. The addition of a clearly stated research aim in the Introduction has significantly improved the framing of the study. The expanded Discussion section now provides a compelling outlook on the translational and clinical implications of the findings, particularly regarding preclinical canine models and their relevance to human immunotherapy.

The authors have also adequately addressed concerns related to T cell functionality markers. While CD69 protein-level assessment was not feasible due to reagent limitations, the explanation is clear and scientifically sound. I agree that functional assays are more appropriate in the context of CAR-expressing cells and appreciate the authors’ transparency and discussion of future directions.

Furthermore, the additional mechanistic insights into the differential effects of antibody clones and substrate formats enhance the rigor of the study. The revisions collectively strengthen the manuscript’s clarity, significance, and potential impact on the field of comparative and translational immunotherapy.

I recommend acceptance of the manuscript in its current form.

Reviewer #2: The author has solved the problem I raised. The quality of revised paper had met the standard of accept.

Reviewer #3: (No Response)

Reviewer #4: Although not all of the suggested experiments were performed, the authors have satisfactorily addressed the majority of the review comments. In my opinion, the revised manuscript has adequately resolved the major concerns and is now suitable for acceptance.

7. PLOS authors have the option to publish the peer review history of their article (what does this mean?). If published, this will include your full peer review and any attached files.

Reviewer #1: No

Reviewer #2: **Yes: **Xiaosheng Liu

Reviewer #3: No

Reviewer #4: **Yes: **Jing Ju

---

## [Editor Report · Acceptance letter]

PONE-D-25-20434R1

PLOS ONE

Dear Dr. Brudno,

I'm pleased to inform you that your manuscript has been deemed suitable for publication in PLOS ONE. Congratulations! Your manuscript is now being handed over to our production team.

Kind regards,

on behalf of

Dr. Xiaosheng Tan

Academic Editor

PLOS ONE